# Genetic dissection identifies Necdin as a driver gene in a mouse model of paternal 15q duplications

Kota Tamada [1,2,3], Keita Fukumoto [1,2], Tsuyoshi Toya[1,4], Nobuhiro Nakai [1,2,3], Janak R. Awasthi[1,5], Shinji Tanaka[6], Shigeo Okabe [6], François Spitz [7], Fumihito Saitow [8], Hidenori Suzuki [8] & Toru Takumi [1,2,3,5✉]

Maternally inherited duplication of chromosome 15q11-q13 (Dup15q) is a pathogenic copy number variation (CNV) associated with autism spectrum disorder (ASD). Recently, paternally derived duplication has also been shown to contribute to the development of ASD. The molecular mechanism underlying paternal Dup15q remains unclear. Here, we conduct genetic and overexpression-based screening and identify *Necdin* (*Ndn*) as a driver gene for paternal Dup15q resulting in the development of ASD-like phenotypes in mice. An excess amount of *Ndn* results in enhanced spine formation and density as well as hyperexcitability of cortical pyramidal neurons. We generate *15q dupΔNdn* mice with a normalized copy number of *Ndn* by excising its one copy from Dup15q mice using a CRISPR-Cas9 system. *15q dupΔNdn* mice do not show ASD-like phenotypes and show dendritic spine dynamics and cortical excitatory-inhibitory balance similar to wild type animals. Our study provides an insight into the role of *Ndn* in paternal 15q duplication and a mouse model of paternal Dup15q syndrome.

[1] RIKEN Brain Science Institute, Wako, Saitama, Japan. [2] Graduate School of Biomedical Sciences, Hiroshima University, Minami, Hiroshima, Japan. [3] Department of Physiology and Cell Biology, Kobe University School of Medicine, Chuo, Kobe, Japan. [4] Graduate School of Pharmaceutical Sciences, Keio University, Minato, Tokyo, Japan. [5] Graduate School of Science and Engineering, Saitama University, Sakura, Saitama, Japan. [6] Department of Cellular Neurobiology, Graduate School of Medicine, The University of Tokyo, Bunkyo, Tokyo, Japan. [7] Department of Human Genetics, University of Chicago, Chicago, IL, USA. [8] Department of Pharmacology, Garduate School of Medicine, Nippon Medical School, Bunkyo, Tokyo, Japan. ✉email: takumit@med.kobe-u.ac.jp

Autism spectrum disorder (ASD) is referred to as a group of neurodevelopmental disorders. The prevalence rate of ASD is now estimated 1 in 59 according to the Centers for Disease Control[1]. Accumulating studies indicate that genetic components are the major contributors to the etiology of ASD, including pathogenic rare variants of a single gene or copy number variations (CNVs)[2–4]. Some CNVs, including 15q11-q13 duplication, are overlapped as shared risk factors for both ASD and schizophrenia, suggesting that the correlation between gene dosage and phenotypes is rather complex[3,5]. Therefore, it is crucial to identify a driver gene within CNV to understand the pathogenesis of these psychiatric disorders.

Chromosome 15q11-q13 duplication (Dup15q syndrome; OMIM# 608636) has been recognized as one of the most cytogenetically abnormal CNVs for ASD, with high prevalence (0.25%)[6]. Chromosome 15q11-q13 is also known to be an imprinting region. In addition to Dup15q, two distinct neurodevelopmental disorders, Prader–Willi syndrome (PWS) and Angelman syndrome (AS), are caused by the deficiency of this paternal and maternal locus, respectively[7]. To understand the pathophysiology of Dup15q syndrome, we previously generated a 15q11-q13 duplication model in mice (15q dup) which has interstitial duplication with 6.3 Mb mouse chromosome 7B-C, corresponding to the human 15q11-q13 locus[8]. Subsequent analyses have revealed multi-dimensional abnormalities in paternal 15q dup mice. Mice with paternally derived duplication displayed ASD-like phenotypes including abnormal social behavior, such as reduced social interaction and behavioral inflexibility[8,9], impaired cortical/cerebellar synaptic functions and morphologies[9–12], and reduced serotonin [5-hydroxytryptoamine (5-HT)] in brain due to hypoactivity and a smaller size of the dorsal raphe nucleus (DRN)[9,13,14]. We also found impaired excitatory/inhibitory (E/I) imbalance in the cerebral cortex[9]. Importantly, the impaired E/I balance and a subset of the abnormal behavior were recovered by increasing 5-HT levels during the postnatal development.

Most individuals with Dup15q syndrome have maternally derived while paternally derived duplication has been considered as non-pathogenic CNVs[15,16]. Therefore, it has been widely accepted that a maternally expressed gene (MEG), UBE3A, is strongly implicated as a driver gene of Dup15q syndrome[16,17]. This idea is supported by an animal study[18]. In contrast, an investigation found individuals with paternally derived duplication also met the criteria for ASD, although its penetrance was estimated at ~20% and the number of cases is still small[19]. Thus, it remains unclear whether paternally expressed genes (PEGs) in 15q11-q13 contribute to the pathogenesis of ASD.

To address the contribution of PEGs in 15q11-q13 for ASD, we first generated a new duplication mouse having a smaller segment than that of the original 15q dup mice and conducted behavioral analyses. We then performed a screening of each PEG to evaluate which PEGs play a role in the altered spine dynamics found in paternal 15q dup mice. Finally, we conducted rescue experiments by excluding the target gene from paternal 15q dup mice.

## Results

### Generation of 1.5 Mb duplication mice encompassing a PWS/AS region.
A previous study suggested that increased expression of Snord115 (also named MBII-52) affected the activity of serotonin 2c receptor (5-HT2cR) in paternal 15q dup mice[8]. To test the functional significance of increased Snord115, we generated the new mice with 1.5 Mb interstitial duplication in chromosome 7B (Fig. 1a and Supplementary Fig. 1), called the PWS/AS locus, by using in vivo chromosome engineering[20,21]. Hprt-Cre mice[22] were crossed with two independent mouse lines containing a loxP

site on the proximal side of Ube3a and the distal side of Snrpn. The target region includes Ube3a, Ube3a antisense RNA (Ube3a-ATS), Snord116, Ipw, Snord115, Snord64, Snord107, Snrpn, and Snurf genes. A total of 105 offspring (F0) were screened and two duplication animals were obtained (hereafter 1.5 Mb Dp/+ mice). The expected recombination was confirmed by conventional genomic PCR, Southern blot (Fig. 1b and Supplementary Fig. 1) and array comparative genomic hybridization (aCGH) (Fig. 1c). Next, we assessed mRNA expressions in the brain. As expected, mRNA expressions of all genes located in the 1.5 Mb duplicated locus were significantly increased without affecting neighbor genes in 1.5 Mb paternally duplicated mice (1.5 Mb patDp/+). PEGs, including Ube3a-ATS, Snord116, Ipw, Snord115, and Snrpn, showed a more than twofold increase in mRNA expression, while Ube3a, an MEG, showed slightly increased expression (Fig. 1d). Because the imprinted expression of Ube3a is neuron specific, it is possible that the increased Ube3a level in 1.5 Mb patDp/+ mice-derived brain tissue might occur due to glial cells. Therefore, we measured the Ube3a expression level in primary neurons and found no statistically significant change between WT and 1.5 Mb patDp/+ mice-derived neurons (Fig. 1e). Therefore, the increased Ube3a expression in brain tissue may reflect that in glial cells.

### Neurochemical analysis in 1.5 Mb duplication mice.
Our previous studies indicated that 15q dup mice exhibit low 5-HT content due to hypoactivity in the DRN[9,13] and amount of serine in the brain[23]. Therefore, we quantified the amounts of 5-HT and serine in the brain of 1.5 Mb Dp/+ mice (Supplementary Fig. 2). Both 1.5 Mb Dp/+ mice with paternally (1.5 Mb patDp/+) and maternally (1.5 Mb matDp/+) inherited duplication showed no significant difference in the amount of 5-HT in six brain regions (Supplementary Fig. 2a; $p > 0.05$, $t$-test). Similarly, the 1.5 Mb duplication did not affect the amount of a metabolite of 5-HT, 5-hydroxyindoleacetic acid (5-HIAA), or of D/L-serine in the brain (Supplementary Fig. 2b, c; $p > 0.05$, $t$-test). Only in the frontal cortex, 1.5 Mb patDp/+ mice had increased 5-HIAA, albeit that the degree of increase seemed very mild (WT, 124.0 ± 6.1 pg/mg tissue weight; 1.5 Mb patDp/+, 155.9 ± 11.9 pg/mg tissue weight), and this alteration differed from that of paternal 15q dup mice.

### Behavioral examination of 1.5 Mb Dp/+ mice.
To evaluate the effect of duplication of the target 1.5 Mb locus on the behavior tasks, we explored the behavioral abnormalities both in 1.5 Mb patDp/+ (Fig. 2a) and 1.5 Mb matDp/+ mice (Supplementary Fig. 3). Consistent with the previous study[8], 1.5 Mb matDp/+ mice did not show behavioral abnormalities found in paternal 15q dup mice though slight increased social interaction time was observed (Supplementary Fig. 3). Taken together with a report of Smith et al., increased single copy of Ube3a is not sufficient to induce ASD-like behaviors in mice[18]. In the open field test, 1.5 Mb patDp/+ mice did not exhibit any significant difference in total distance traveled (Fig. 2b) or time spent in the center area of the open field (Fig. 2c), compared to WT mice (two-way repeated measures ANOVA; genotype effect, $p > 0.05$). Moreover, the anxiety index[24,25], the ratio of distance in the center area:total distance, was similar between 1.5 Mb patDp/+ and WT mice (Fig. 2d, $p > 0.05$). Next, we tested sociability of 1.5 Mb patDp/+ mice by the reciprocal social interaction test (Fig. 2e). Many mouse models for ASDs including CNV or rare variants associated with ASD often show less time spent in interaction between two mice[26,27]. Interaction time in 1.5 Mb patDp/+ mice was similar to that of WT mice ($p > 0.05$). Finally, we evaluated reversal learning ability by using a Barnes maze[28] (Fig. 2f–k). The spatial learning task was conducted for 6 days followed by the

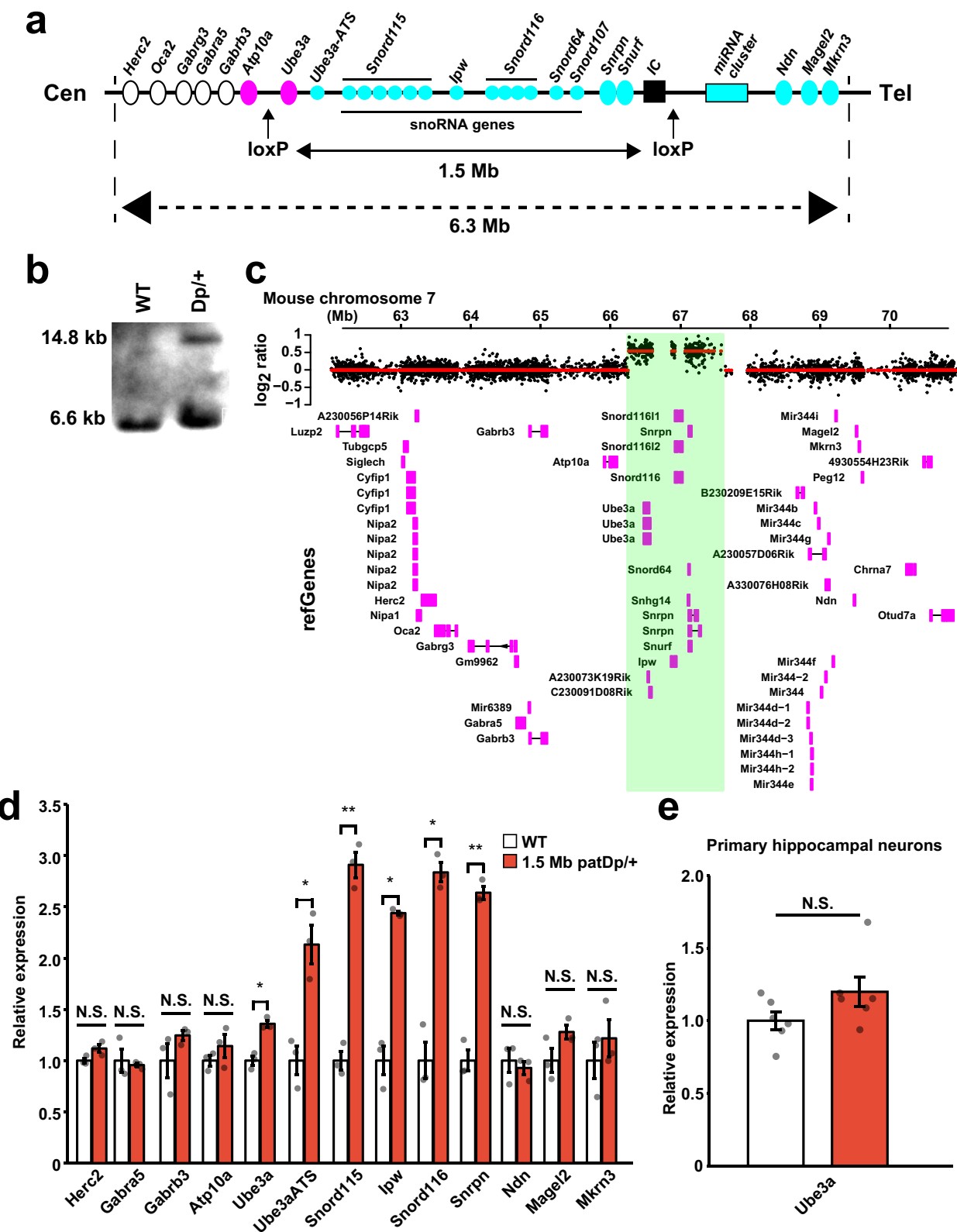

reversal learning task for 4 days (Fig. 2f, g). During the training, the number of errors to reach the correct hole by *1.5 Mb patDp/+* mice was similar as that in WT mice both in spatial and reversal learning tasks (Fig. 2g, $p > 0.05$; both genotype effect and interaction with time). To ensure the spatial learning ability in *1.5 Mb patDp/+* mice, a probe trial was conducted after 6 days learning (Fig. 2h, i). Similar to WT mice, *1.5 Mb patDp/+* mice spent more

time in the correct quadrant (TA) than other quadrants of field (Fig. 2h, i; $p < 0.01$, Dunnet's test). Furthermore, *1.5 Mb patDp/+* mice learnt the new target position in the probe trial 4 days after reversal learning (Fig. 2j, k). Taken together, these data implicated PEGs in the 1.5 Mb duplicated locus containing *Ube3a-ATS*, *Snord115*, *Ipw*, *Snord116*, *Snord64*, *Snord107*, *Snrpn*, and *Snurf* do not have a critical role for ASD-like behaviors.

**Fig. 1 Generation of chromosome duplication mice encompassing 1.5 Mb in PWS/AS locus. a** A schematic map of mouse chromosome 7 PWS/AS locus and target for duplication (Cen centromere, Tel telomere). The original 6.3 Mb duplication mice (*15q dup*) have an interstitial duplication from *Herc2* to *Mkrn3*, while 1.5 Mb duplication mice have this from *Ube3a* to upstream of the imprinting center (IC; black box). Blue, pink, and white circles indicate paternally, maternally and biallelically expressed genes, respectively. **b** Southern blot analysis in WT and *1.5 Mb Dp/+* mice. The band size of the Dp allele is ~14.8 Kb. At least, two independent mice were used and similar results were obtained. **c** Oligo nucleotide-based comparative genomic hybridization (aCGH). WT genome DNA was used as a reference DNA and plotted as the log2-transformed signal ratio of *1.5 Mb Dp/+* to reference. The base-pair position along mouse chromosome 7 was described horizontally (mm9). A red line indicates the estimated mean of each segment. A green shaded region is an expected duplicated locus. **d** mRNA expressions in hippocampal tissue of *1.5 Mb patDp/+* mice were measured by RT-qPCR ($N = 3$ biologically independent mice for each genotype). *Ube3a*: $p = 0.0044$, *Ube3a-ATS*: $p = 0.0099$, *Snord115*: $p = 0.0004$, *Ipw*: $p = 0.0083$, *Snord116*: $p = 0.0025$, *Snrpn*: $p = 0.0004$. **e** *Ube3a* mRNA expression in primary hippocampal neurons (DIV 7) measured by RT-qPCR ($N = 6$ biologically independent embryos for each genotype). Each value was normalized by *Gapdh* as an internal control and described as a relative value of the mean of WT expressions. *$p < 0.05$, **$p < 0.01$ (*t*-test after correction for multiple testing by Benjamini–Hochberg procedure). Data are represented as mean ± s.e.m. N.S. not statistically significant.

**Screening of PEGs in 15q11-q13 for spine phenotypes**. From these behavioral and neurochemical data for *1.5 Mb patDp/+* mice described above, the remaining PEGs, including *Ndn*, *Magel2*, and *Mkrn3*, were thought to be the next candidates for a driver gene. To identify which gene plays a critical role in ASD phenotypes, we performed a screening for enhanced synaptic remodeling, which is a common feature in ASD model mice, by in vivo imaging using a two-photon microcopy[12]. Indeed, ASD is classified as synaptopathy[4,29]. We introduced enhanced green fluorescent protein (EGFP) with each target gene to layer II/III pyramidal neurons in the somatosensory cortex by in utero electroporation, and then measured the spine turnover rate for 2 days in vivo (Fig. 3a). The newly formed spines were drastically increased for 2 days following introduction of the *Ndn* gene (Fig. 3b). As a result of quantification, exogenous *Ndn* expression significantly increased the spine formation rate and density while the elimination rate appeared to be increased, but not significantly so (Fig. 3c–e, $p < 0.01$, Dunnett's test). The overexpression level of *Ndn* was verified by immunohistochemistory, and it showed about threefold increase compared to non-transfected neurons (Supplementary Fig. 4). Among other genes, *Magel2* slightly enhanced both the formation rate and density and *Snrpn* enhanced the elimination rate. However, the effect size of *Magel2* (Hedges' $g = 2.3$ for formation rate and 3.6 for density against control) was smaller than that of *Ndn* (5.6 for formation rate and 6.9 for density against control). Moreover, the spine density of *Snrpn* was not statistically different from control neurons (Fig. 3e). Therefore, this analysis revealed *Ndn* is a primary regulator in the proper development of dendritic spines in 15q11-q13 PEGs. To date, however, its function for dendritic spines has remained poorly understood. To investigate the role of *Ndn* for dendritic spines in more detail, we examined the morphology of dendritic spines in neurons with *Ndn* overexpression (Fig. 3f, g). Dendritic spines are classified into four categories by maturation stage[30]. Filopodia are thought to be a precursor form of dendritic spines and play important roles at the initial stages of synaptogenesis. Neuronal activity induces filopodia to form stubby, thin and mushroom-type spines which contain scaffold proteins and receptors[31]. This classification analysis revealed that *Ndn*-overexpressing neurons had increased density of filopodia and stubby-type spines (Fig. 3f). Thus, the proportion of immature filopodia-type spines and mature mushroom-type ones was increased and decreased, respectively [Control: 16.9 ± 1.4% (filopodia), 23.5 ± 1.2% (stubby), 39.6 ± 1.7% (thin), 20.0 ± 1.3% (mushroom); *Ndn*: 34.0 ± 1.7% (filopodia), 24.7 ± 1.5% (stubby), 28.5 ± 1.7% (thin), 12.8 ± 1.5% (mushroom)] (Fig. 3g, chi-squared test; $p < 0.05$). These results indicate that *Ndn* regulates the number and maturation of dendritic spines in the cerebral cortex.

**Loss of function of Ndn induces the reduction of spine formation**. We next investigated whether loss of function of *Ndn* causes the opposite effect on spines as overexpression. We validated the spine turnover rate in *Ndn* KO mice[32] by in vivo imaging using two-photon microscopy. In contrast to *Ndn* overexpression, *Ndn* KO mice showed a significantly lower spine formation rate but an elimination rate comparable to that of WT mice (Supplementary Fig. 5a–c). To obtain further molecular insight, we conducted a similar analysis with overexpression of a deletion construct of *Ndn*. *Ndn* gene is a member of melanoma antigen gene (MAGE) family, although the function of this domain remains unknown. NDN has been considered an adapter protein, and it is therefore important to identify the interaction partner that regulates spine formation. The N-terminal deletion form of NDN lacks binding affinities for some proteins and transcriptional activity[33]. Following previous reports, we prepared the N-terminal deletion construct of *Ndn* (NDNΔN) and investigated its overexpression effect on spine dynamics (Supplementary Fig. 6a, b). Compared to intact NDN-induced neurons, NDNΔN-induced neurons exhibited significantly decreased levels in both formation rate and density (Supplementary Fig. 6c–e), but no statistical difference in elimination rate. These results suggest that the N-terminal domain of NDN is partly required for regulation of dendritic spines; in other words, the target proteins that bind to the N-terminal of NDN may play a role in spine formation in the cerebral cortex.

**Overexpression of Ndn induces hyperexcitability in cortical neurons**. To determine the effect of *Ndn* overexpression on synaptic strength and excitability, we conducted whole-cell recording in cortical slices. The intrinsic excitability of layer II/III pyramidal neurons in the original paternal *15q dup* mice was higher than that of WT[9]. We introduced an *Ndn*-overexpressing plasmid with EGFP to the somatosensory cortex by in utero electroporation and prepared acute brain slices from control (i.e., empty vector was introduced) and *Ndn*-overexpressing mice. *Ndn*-overexpressed neurons showed decreased amplitude, a lower frequency of inter-event interval of miniature excitatory postsynaptic currents, and increased membrane input resistance, whereas the resting membrane potential and the cell capacitance were equivalent to those of control neurons (Fig. 4a–e). Importantly, *Ndn*-overexpressed neurons fired with higher frequency than control neurons on current step injections (Fig. 4f, g; two-way ANOVA, main effect of overexpression, $p < 0.01$), indicating that *Ndn* regulates intrinsic neuronal excitability. This electrophysiological characteristic is similar to that of neurons of paternal *15q dup* mice shown in our previous study[9]. Collectively, these findings indicate that increased expression of *Ndn* contributes to the hyperexcitability in cortical pyramidal neurons.

**Generation of mice with single copy removal of Ndn from 15q dup mice**. It is possible that the overexpression of *Ndn* achieved

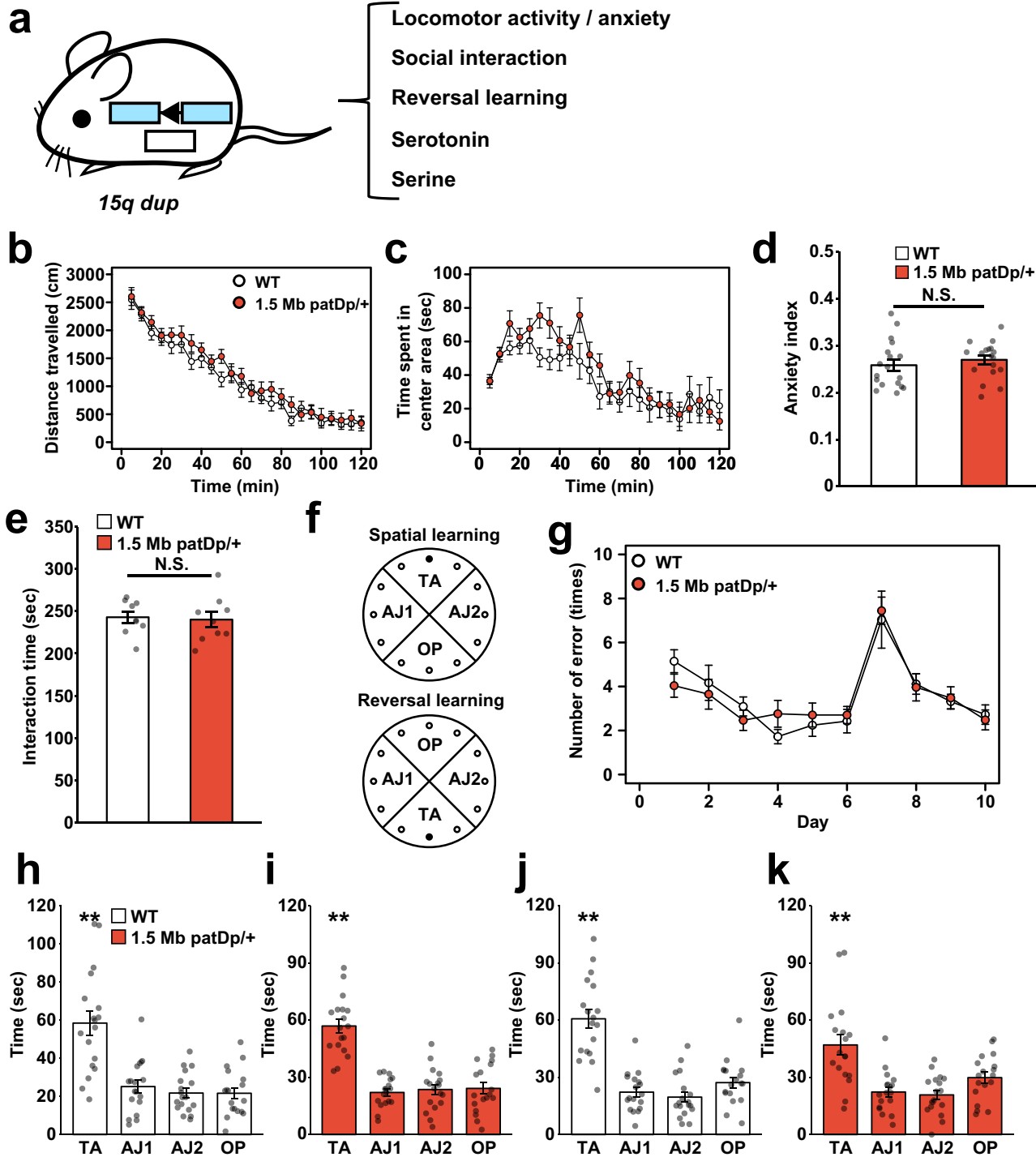

**Fig. 2 Paternal duplication mice of 1.5 Mb do not replicate abnormal behaviors found in paternal *15q dup* mice. a** A schematic of phenotypes of paternal *15q dup* mice based on published studies. **b–d** Open field test in *1.5 Mb patDp/+* mice. Mean values in 5-min segments are plotted including **b** locomotor activity and **c** time spent in the center area. **d** Anxiety index is calculated as distance in the center area divided by total distance during the first 30 min. $N = 18$ biologically independent mice for each genotype. **e** Reciprocal social interaction test. The interaction time for each pair of mice is shown. $N = 9$ pairs of mice for each genotype. **f–k** Spatial and reversal learning memory was evaluated by the Barnes maze test. **f** A schematic view of two learning paradigms. TA target quadrant, AJ1/2 adjacent quadrant, OP opposite quadrant. A black filled circle indicates the correct hole with an escapable box under the hole. **g** The number of errors until reaching the correct target is shown. Spatial and reversal learning was conducted on days 1–6 and 7–10, respectively. Results of probe test on day 6 (**h**: WT, **i**:*1.5 Mb patDp/+*). WT: TA vs. AJ1/AJ2/OP: $p < 0.0001$, *1.5 Mb patDp/+*: TA vs. AJ1/AJ2/OP: $p < 0.0001$. **$p < 0.01$ (Dunnett's test). **j**, **k** Results of the reversal probe test on day 10. $N = 18$ biologically independent mice for each genotype. WT: TA vs. AJ1/AJ2/OP: $p < 0.0001$, *1.5 Mb patDp/+*: TA vs. AJ1/AJ2: $p < 0.0001$, TA vs. OP: $p = 0.0023$. **$p < 0.01$ (Dunnett's test). Data are represented as mean ± s.e.m. N.S. not statistically significant.

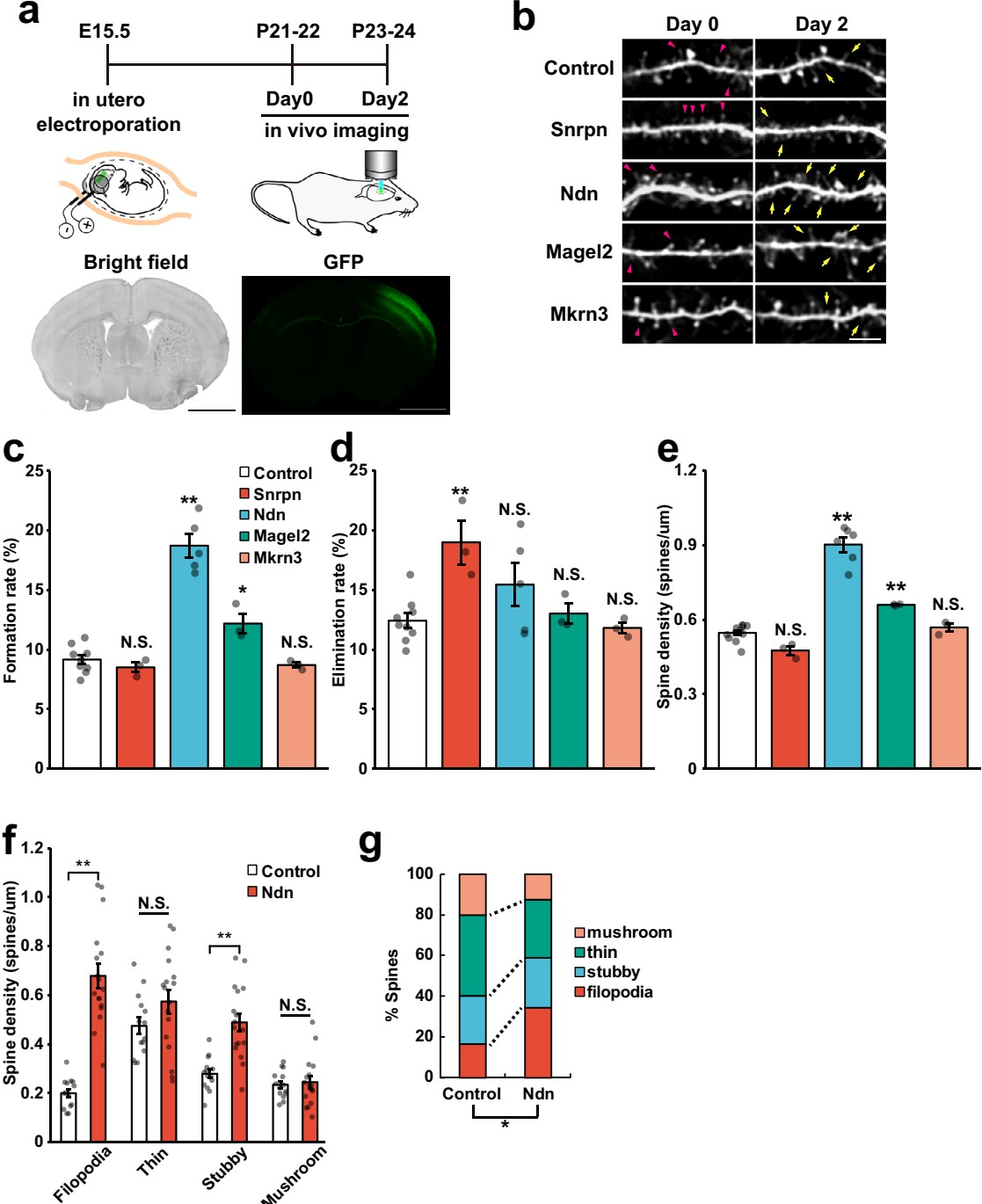

**Fig. 3 *Ndn* regulates formation and maturation of dendritic spines. a** (Upper): Experimental schedule for in vivo imaging. (Bottom): images of somatosensory cortex electroporated with GFP plasmid at postnatal 3 weeks (P21-24). **b** Representative images of dendritic spines by in vivo time-lapse imaging. Arrows and arrowheads indicate newly formed and eliminated spines within 2 days, respectively. Similar images were obtained in the independent mice (at least more than 3). **c, d** Quantification of spine dynamics by introducing each 15q11-13 PEG. **c** Spine formation rate and **d** elimination rate are shown. $N = 9$ biologically independent mice (control), 5 (*Ndn*) and 3 (*Mkrn3, Magel2,* and *Snrpn*). Formation rate: *Snrpn*: $p = 0.9205$, *Ndn*: $p < 0.0001$, *Magel2*: $p = 0.0197$, *Mkrn3*: $p = 0.9764$. Elimination rate: *Snrpn*: $p = 0.0046$, *Ndn*: $p = 0.1644$, *Magel2*: $p = 0.9927$, *Mkrn3*: $p = 0.9926$. **e** The density of spines was evaluated by immunohistochemistry using the anti-GFP antibody. $N = 12$ biologically independent mice (control), 6 (*Ndn*) and 3 (*Mkrn3, Magel2,* and *Snrpn*). Spine density: *Snrpn*: $p = 0.0688$, *Ndn*: $p < 0.0001$, *Magel2*: $p = 0.0022$, *Mkrn3*: $p = 0.8872$. **f, g** Classification of each spine. **f** Spine density in each class. Filopodia: $p < 0.0001$, Thin: $p = 0.1$, Stubby: $p < 0.0001$, Mushroom: $p = 0.720$. **g** Proportion of each classified spine. $N = 14$ (control) and 17 (*Ndn*) of the dendrites from biologically independent three mice. **$p < 0.01$, *$p < 0.05$ (**c–e**: Dunnett's test, **f**: *t*-test after correction for multiple testing by Benjamini–Hochberg procedure, **g**: chi-squared test). Scale bar for **a**, 500 μm; **b**, 5 μm. Data are represented as mean ± s.e.m. N.S. not statistically significant.

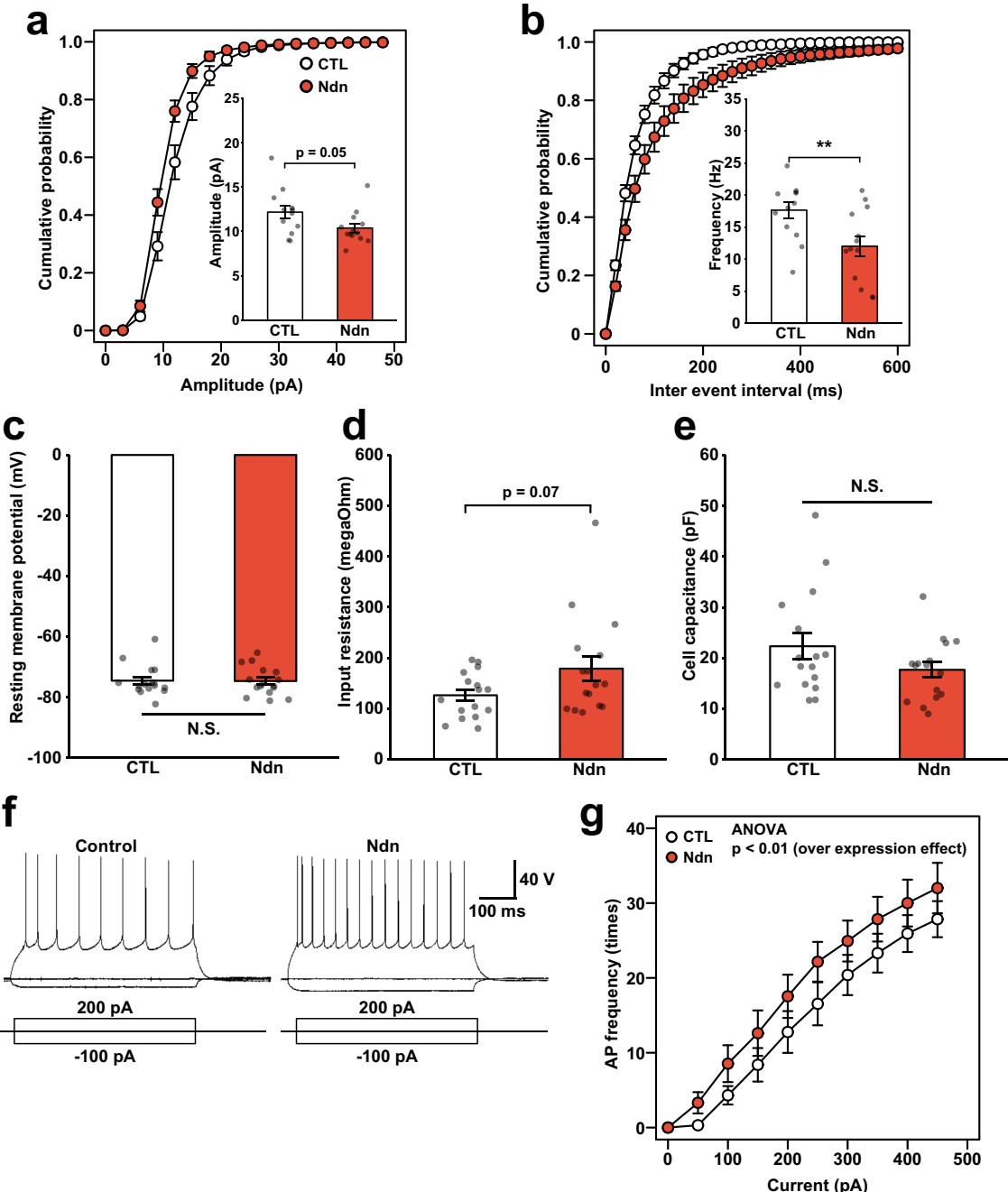

**Fig. 4 Exogenous *Ndn* enhances neuronal excitability in cortical pyramidal neurons. a**, **b** mEPSC recording in somatosensory cortex layer II/III pyramidal neurons overexpressing *Ndn*. Cumulative distributions and mean of mEPSC **a** amplitude, **b** inter-event intervals. Frequency: CTL vs. *Ndn*, $p = 0.01$ (*t*-test). $N = 13$ neurons from six biologically independent mice for each group. Measurement of **c** resting membrane potential, **d** input resistance, **e** cell capacitance. $N = 16$ neurons from six mice for each genotype. **f**, **g** Action potential frequencies in *Ndn*-overexpressing neurons. **f** Representative traces of action potentials with injected currents. **g** The number of APs elicited by sequential current injections from 0 to 450 pA. $N = 13$ neurons from biologically independent six mice for each genotype. **g** **\*\***$p < 0.01$ (two-way ANOVA; effect of overexpression: $p = 0.0004$, applied current $p < 0.0001$, interaction $p = 0.9945$). Data are represented as mean ± s.e.m. N.S. not statistically significant.

using in utero electroporation with a strong artificial promoter may not reflect physiological conditions. Therefore, we examined whether removing a single copy of *Ndn* from original paternal *15q dup* mice is sufficient to restore the abnormal phenotypes (Fig. 5a; named *15q dupΔNdn*). To produce this new mouse, we applied the CRISPR-Cas9 system. We designed two guide RNAs that encompassed a whole *Ndn* coding region (Supplementary Fig. 7a; sgNdnProx and sgNdnDist) and introduced these guide RNAs with a *spCas9*-expressing plasmid to Neuro2a cell lines. We

were able to detect the desired genomic deletion only when both guide RNAs were transfected (Supplementary Fig. 7b). We then performed cytoplasmic injection of *Cas9* mRNA with two guide RNAs to fertilized eggs obtained by crossing paternal *15q dup* male mice and WT female mice. Twenty-nine pups with a duplication allele were born, of which 12 pups had one or two deletions in the *Ndn* locus, as determined by conventional genomic PCR. As further validation, we evaluated the target deletion by Southern blot and genomic PCR in mice

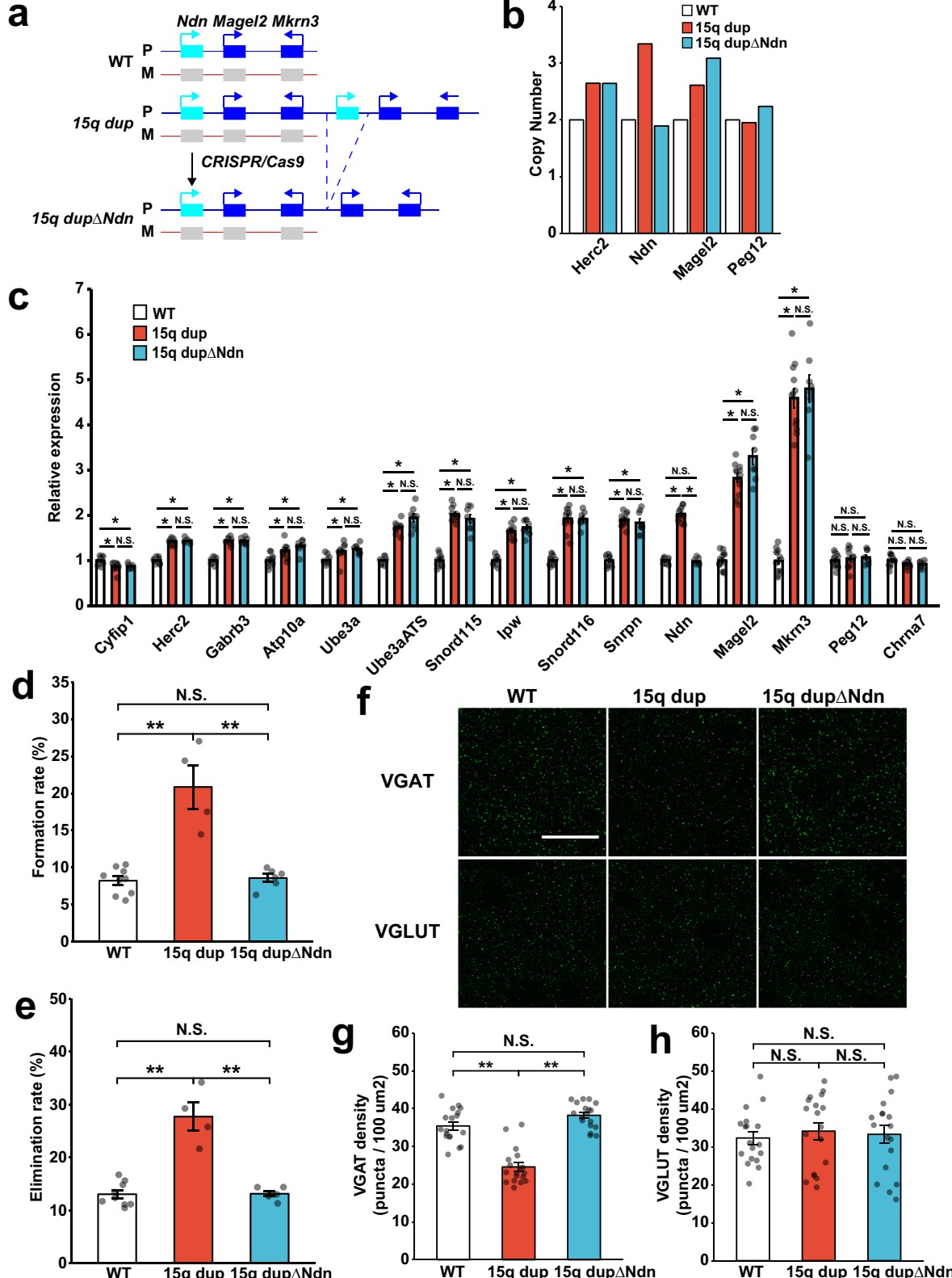

(Supplementary Fig. 7c–e). By genomic PCR and Southern blot, we could not distinguish a 1 or 2 copy deletion from *15q dup* mice having three copies of *Ndn*. To clarify the exact genomic copy number of *Ndn* in the target mice, we performed droplet digital PCR (ddPCR) with genomic DNA. The copy number of *Ndn* in *15q dupΔNdn* mice was comparable to that of WT mice (two copies), while other genes included in a duplicated allele,

*Herc2* and *Magel2*, showed around three copies both in *15q dup* and *15q dupΔNdn* mice (Fig. 5b). Moreover, we compared the gene expression level in the brain of paternal *15q dupΔNdn* mice and found that the expression level of *Ndn* was also normalized to that of WT mice (Fig. 5c). Taken together, these findings indicated that we succeeded in generating *15q dupΔNdn* mice that contained a normalized genomic copy number and gene

**Fig. 5 Normalized genomic copy number of *Ndn* ameliorates enhanced dendritic spine turnover and cortical excitatory/inhibitory imbalance found in paternal *15q dup* mice. a** Schema of generation of *15q dup∆Ndn* mice. **b** Measurement of genomic copy number by digital quantitative PCR. The copy number of WT set to 2. **c** mRNA expression in the frontal cortex of paternal *15q dup* and *15q dup∆Ndn* mice was measured by RT-qPCR. Each value was normalized by *Gapdh* as an internal control and described as a relative value of WT expression (*N* = 14, 12, 8 biologically independent mice for WT, *15q dup*, and *15q dup∆Ndn*, respectively). All comparisons are shown in Source Data. **d**, **e** Quantification of spine dynamics in *15q dup* and *15q dup∆Ndn* mice crossed with *Thy1-YFP-H* transgenic mice. **d** Spine formation and **e** elimination rate is shown. *N* = 9 (WT), 4 (*15q dup*), 6 (*15q dup∆Ndn*) biologically independent mice. Spine formation rate: WT vs. *15q dup*: *p* < 0.0001, WT vs. *15q dup∆Ndn*: *p* = 0.971, *15q dup* vs. *15q dup∆Ndn*: *p* < 0.0001. Elimination rate: WT vs. *15q dup*: *p* < 0.0001, WT vs. *15q dup∆Ndn*: *p* = 0.998, *15q dup* vs. *15q dup∆Ndn*: *p* < 0.0001. Cortical E/I balance evaluated by immunohistochemistry for (**f**; upper panel, **g**) VGAT and (**f**; lower panel, **h**) VGLUT in S1BF layer II/III. *N* = 18 images from biologically independent 3 mice for each genotype. VGAT: WT vs. *15q dup*: *p* < 0.0001, WT vs. *15q dup∆Ndn*: *p* = 0.115, *15q dup* vs. *15q dup∆Ndn*: *p* < 0.0001. VGLUT: WT vs. *15q dup*: *p* = 0.82, WT vs. *15q dup∆Ndn*: *p* = 0.945, *15q dup* vs. *15q dup∆Ndn*: *p* = 0.957. **c**; *t*-test after correction for multiple testing by Benjamini–Hochberg procedure. **p* < 0.01. **d**, **e**, **g**, **h**; Tukey–Kramer comparison test ***p* < 0.01. Scale bar in **f**, 10 μm. Data are represented as mean ± s.e.m. N.S. not statistically significant.

expression of *Ndn* but retained the increased expression of other duplicated genes.

**Single copy removal of Ndn from paternal 15q dup mice restores alteration in dendritic spines and cortical E/I imbalance found in paternal 15q dup mice.** To confirm the contribution of *Ndn* to dendritic spine dynamics, we visualized dendritic spines in cortical neurons of paternal *15q dup∆Ndn* mice by crossing with *Thy1-YFP-H* mice and comparing their spine turnover rate by in vivo imaging using a two-photon microscopy. The aberrant enhanced spine formation rate in paternal *15q dup* mice was completely abolished in paternal *15q dup∆Ndn* mice and its turnover rate was equivalent to that of WT (Fig. 5d). Surprisingly, the spine elimination rate was also normalized in paternal *15q dup∆Ndn* mice (Fig. 5e). Cortical E/I imbalance has been frequently observed in animal models of ASD and their correction might be one of the key points in normalizing ASD-like phenotypes[34–36]. In our previous study, paternal *15q dup* mice showed a decreased frequency of inhibitory synaptic input and decreased numbers of an inhibitory synaptic vesicle marker (VGAT)[9]. We were able to replicate the reduced VGAT density in the somatosensory cortex of paternal *15q dup* mice (Fig. 5f, g). The number of VGAT-immunopositive puncta in paternal *15q dup∆Ndn* mice was higher than that in paternal *15q dup* mice (Fig. 5g), but similar to that in WT mice. The number of excitatory synaptic marker (VGLUT) was similar among all genotypes (Fig. 5h). These data indicated *Ndn* has a predominant role in the proper development of cortical circuits among mouse homologous genes in 15q11-q13.

**Paternal 15q dup∆Ndn mice are recovered from abnormal ASD-like behaviors.** We investigated whether *15q dup∆Ndn* mice could be rescued from the ASD-like behaviors found in paternal *15q dup* mice, such as high anxiety in a novel environment in the open field test, decreased social interaction (based on reciprocal social interaction and three-chambered social interaction tests), and impaired reversal learning[8,9,13,23]. The open field test showed no significant difference between WT and paternal *15q dup∆Ndn* mice in exploratory activity (Fig. 6a), time spent in the center area (Fig. 6b), or anxiety index, calculated by distance in the center area divided by total distance traveled (Fig. 6c). We then performed two types of social interaction tests, the reciprocal social interaction test and three-chambered social interaction test, to verify the sociability for paternal *15q dup∆Ndn* mice[37]. In the reciprocal social interaction test, the approach time did not differ between paternal *15q dup∆Ndn* and WT mice (Fig. 6d). Supporting this finding, paternal *15q dup∆Ndn* mice did not show a defect in sociability for a stranger mouse in the three-chambered test as seen in littermate WT mice (Fig. 6e, f). We also examined the effect of *Ndn* removal on reversal learning ability by the

Barnes maze test. A probe trial after spatial learning phase revealed that time spent in the correct quadrant was significantly increased compared to other quadrants both in WT and paternal *15q dup∆Ndn* mice (Fig. 6g, h). Similarly, both genotype mice showed longer time spent in a new target quadrant in a probe trial after the reversal learning phase (Fig. 6i, j). These behavioral analyses indicated that an increase in *Ndn* expression is required for exhibiting abnormal ASD-like behaviors in paternal *15q dup* mice.

In addition to impaired social interaction, ASDs are also characterized by another trait, impaired social communication. To reveal the communication alteration in paternal *15q dup∆Ndn* mice, we conducted ultrasonic vocalization (USV) tests under two distinct paradigms involving USVs of pups induced by maternal isolation and female-induced male USVs in a resident-intruder paradigm[38]. Based on our previous studies, paternal *15q dup* mice emitted an increased number of USVs on postnatal day 7 and fewer calls in female-induced male USVs in the adult stage[8]. Both paternal *15q dup* and *15q dup∆Ndn* pups exhibited more USVs than WT (Supplementary Fig. 8a). Moreover, paternal *15q dup∆Ndn* adult male mice showed fewer USVs than WT mice when exposed to an estrus female mouse (Supplementary Fig. 8b). Therefore, these data suggest that the abnormal social communication found in paternal *15q dup* mice was not caused by an effect of increased expression of *Ndn*.

Finally, we evaluated the membrane properties and spontaneous EPSC (sEPSC) in the raphe nuclei of paternal *15q dup∆Ndn* mice. Our previous study revealed paternal *15q dup* mice exhibited hyperpolarized resting membrane potentials of 5-HT neurons in the DRN and decreased sEPSC amplitude whereas sEPSC frequencies showed no change[9]. This analysis revealed that paternal *15q dup∆Ndn* mice have a tendency to decrease of sEPSC amplitude both in the lateral wing of DRN and ventromedial DRN, similar to paternal *15q dup* mice (Supplementary Fig. 9a, c; *p* = 0.07, Mann–Whitney *U* test), while the frequency of sEPSC in paternal *15q dup∆Ndn* mice was comparable to that of WT mice (Supplementary Fig. 9b, d). Interestingly, hyperpolarized resting membrane potential found in paternal *15q dup* mice was not observed in paternal *15q dup∆Ndn* mice (Supplementary Fig. 9e, f), suggesting the rescued intrinsic excitability of 5-HT neurons in paternal *15q dup∆Ndn* mice.

## Discussion

Our screening revealed that one of the PEGs in the 15q11-q13 locus, *Ndn*, has a critical role in developing ASD-related phenotypes seen in paternal *15q dup* mice. The exogenous induction of *Ndn* caused an aberrant increase in dendritic spines (mostly immature spines) and neuronal hyperexcitability that are reminiscent of the phenotypes in paternal *15q dup* mice. Single copy removal of *Ndn* from *15q dup* mice was sufficient to alleviate the

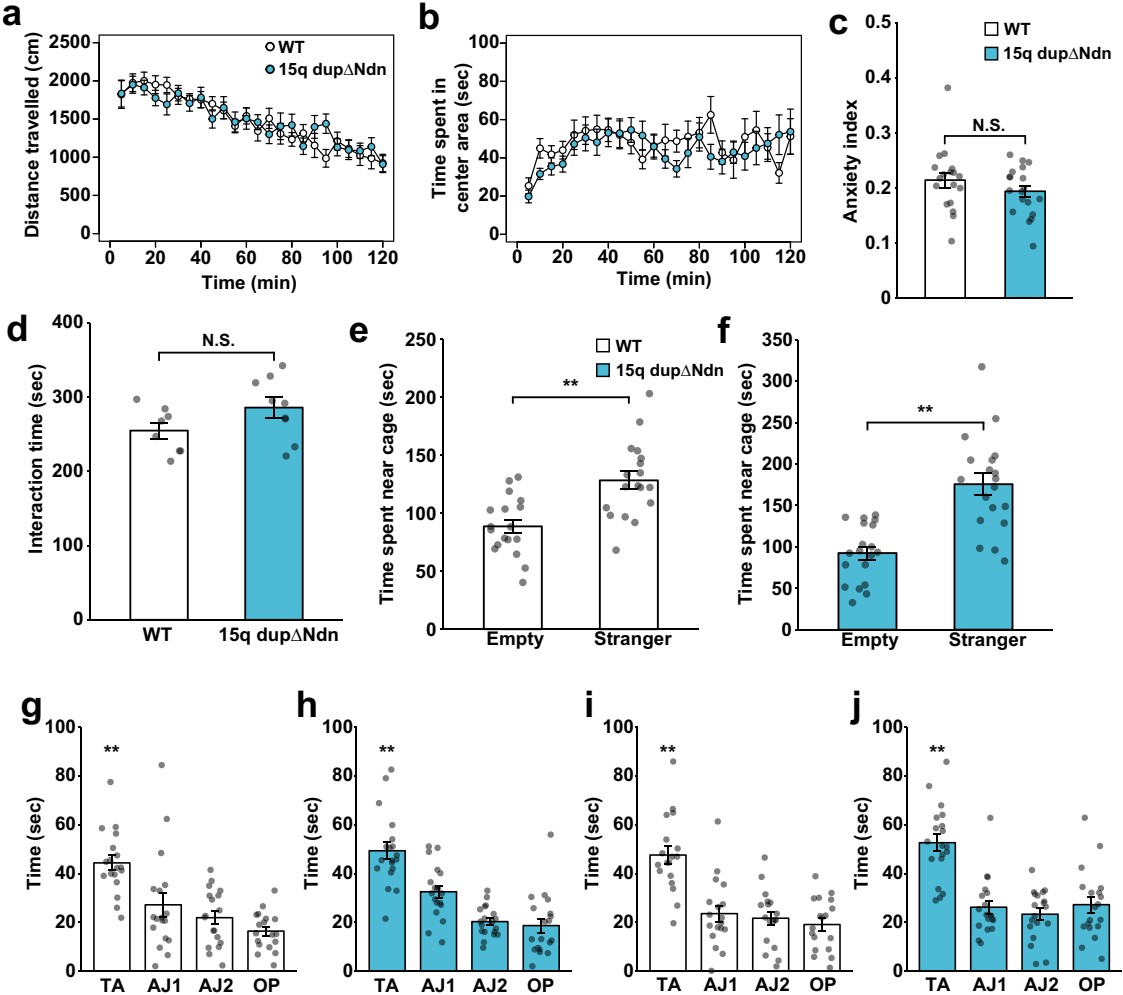

**Fig. 6 Paternal *15q dupΔNdn* mice are recovered from abnormal behaviors found in paternal *15q dup* mice. a–c** Open field test. **a** total distance traveled, **b** time spent in the center area, and **c** anxiety index. *N* = 18, 19 biologically independent mice for WT and *15q dupΔNdn* mice. **d** Reciprocal social interaction test. The time spent in social interaction for each pair of mice is shown. *N* = 8 and 9 pairs for WT and *15q dupΔNdn* mice, respectively. **e**, **f** Three-chambered social interaction test. (Left) WT *N* = 18, (Right) *15q dupΔNdn N* = 19 biologically independent mice. WT: Empty vs. Stranger: *p* = 0.0004, *15q dupΔNdn*: Empty vs. Stranger: *p* = 0.0002. **\*\****p* < 0.01 (paired *t*-test). **g**, **h** Results of the probe test after spatial learning training in the Barnes maze test. WT: TA vs. AJ1: *p* = 0.0013, TA vs. AJ2/OP: *p* < 0.0001, *15q dupΔNdn*: TA vs. AJ1/AJ2/OP: *p* < 0.0001. **\*\****p* < 0.01 (Dunnett's test). **i**, **j** Results of the reversal probe test. *N* = 18 for WT, 19 for *15q dupΔNdn* biologically independent mice. WT: TA vs. AJ1/AJ2/OP: *p* < 0.0001, *15q dupΔNdn*: TA vs. AJ1/AJ2/OP: *p* < 0.0001. **\*\****p* < 0.01 (Dunnett's test). Data are represented as mean ± s.e.m. N.S. not statistically significant.

multi-dimensional abnormalities found in paternal *15q dup* mice. These findings indicate that the appropriate expression of *Ndn* is required for the proper development of cortical circuits and prevention of ASD-like behaviors.

Dup15q is recognized as one of the best-known CNVs in ASD. Due to the predominant number of individuals with its maternally derived duplication, PEGs have not been well explored as an ASD risk and have not been considered pathogenic to date. However, this does not mean that PEGs in 15q11-q13 are not related to ASD. First, the general population frequency of maternal duplication is about double that of paternal duplication[19]. The relatively rare occurrence of paternal duplication may have resulted in their being overlooked in previous studies. Moreover, while a few reports have shown atypical duplication in ASD, these have only included the *NDN*, *MAGEL2*, and *MKRN3* genes, and excluded *UBE3A*[39,40]. Therefore, we tried to address which gene is critical for developing ASD in 15q11-q13 using a model mouse.

We are the first to identify *Ndn* as a critical gene in 15q11-q13 for developing ASD-like behaviors in mice. We initially focused

on *Snord115* because our previous study showed that the increased expression of *Snord115* in paternal *15q dup* mice affected RNA editing of 5-HT2cR and induced altered the intracellular Ca$^{2+}$ response elicited by its agonist[8]. We generated 1.5 Mb duplication mice encompassing *Ube3a*, *Ube3a-ATS*, *Snord116*, *Ipw*, *Snord115*, *Snrpn*, *Snurf* and several functionally uncharacterized snoRNAs. Paternally derived duplication of this region did not show the behavioral abnormalities found in original paternal *15q dup* mice including decreased social interaction, novelty-induced anxiety and deficit in reversal learning. Therefore, the gene dosage of PEGs in this region was not critical for the phenotypes. We then performed a second screening using a spine phenotype, given that excess turnover of dendritic spines is a common endophenotype in ASD model mice[12]. Among four PEGs, we found that overexpression of *Ndn* dramatically enhanced dendritic spine formation rate. In contrast, *Snrpn*, *Magel2*, and *Mkrn3* did not cause an enhanced turnover rate or a milder effect than *Ndn*. Finally, we generated mice having a normalized copy number of *Ndn* in paternal *15q dup*, named *15q dupΔNdn* mice. Surprisingly, paternal *15q dupΔNdn* mice were

recovered from not only the enhanced spine turnover rate but also the cortical E/I imbalance, and several impaired behaviors found in paternal *15q dup* mice. These data solved what had hitherto been a puzzle in 15q11-q13 duplication, and expanded our knowledge for future therapeutic intervention.

**Causative genes for developing 15q11-q13 duplication syndrome**. Early studies reported that maternally derived duplication was found in autism, not paternally derived duplication[15,16]. Subsequent reports of human genetics and a mouse model strongly indicated that an MEG, *UBE3A*, has critical roles in Dup15q syndrome[18,41]. Nevertheless, paternal-derived duplication has also been identified in developmental disorders, albeit that the number of individuals is still limited and its penetrance is incomplete[42–45]. Isles et al. first investigated a relatively large cohort of 15q11-q13 interstitial duplication[19], and reported that paternal duplication might lead to an increased risk of developing developmental delay/ASD/multiple congenital anomalies. Furthermore, a few studies newly reported atypical duplication, including the *NDN*, *MAGEL2*, and *MKRN3* genes only[39,40]. These studies may suggest the relevance of PEG and Dup15q syndrome in addition to *UBE3A* though paternal and maternal Dup15q syndrome may be different syndrome.

**Control of developmental spine turnover, neuronal excitability, and cortical E/I balance by Ndn**. Our screening revealed that overexpression of *Ndn* drastically enhanced the spine formation rate and these spines were largely immature filopodia-type spines (Fig. 3). In contrast, *Ndn* KO mice or N-terminal deletion form of NDN showed a reduced spine turnover rate (Supplementary Figs. 5 and 6). Therefore, we concluded that *Ndn* regulates the number and maturation of cortical dendritic spines in a dosage-dependent manner. In support of this idea, paternal *15q dup* mice also had an increased proportion of immature spines[11]. This alteration in spines is important in the pathogenesis of ASD because other ASD model mice such as *Fmr1* KO mice exhibited similar abnormality of spines with several behavioral defects (reviewed in[46]). Although clarifying how these altered spines affect behaviors remains challenging, the microcircuit level in the cerebral cortex should at the least be perturbed. Indeed, *Ndn*-overexpressed neurons exhibited hyperactivity (Fig. 4), and cortical E/I balance in paternal *15q dup* mice was altered whereas paternal *15q dupΔNdn* mice were comparable to WT mice (Fig. 5).

We also found that *Magel2* and *Snrpn* overexpression caused increased spine formation/density and decreased elimination rate, respectively (Fig. 3). With regard to *Magel2*, *Ndn* and *Magel2* may cooperatively regulate spine turnover. Previous reports showed that NDN binds to MAGEL2 and regulates leptin receptor sorting/degradation or axonal outgrowth through a ubiquitin-dependent pathway[47,48]. Indeed, restoring only *Ndn* from paternal *15q dup* mice could not rescue abnormalities in social communication (Supplementary Fig. 8). Thus, a similar cooperative function of *Ndn* with *Magel2* in spine formation or cortical E/I balance might be an interesting perspective. Regarding *Snrpn*, the reduction in spine elimination rate was not supported by the spine density (Fig. 3d, e). Li et al. recently reported that SNRPN regulated dendritic spine density[49], and that overexpression of *Snrpn* increased the number of dendritic spines via *Nr4a1*. In contrast, we found no significant difference in spine density by overexpression of *Snrpn* (Fig. 3e). This inconsistency is likely due to the difference in quantification method. Li et al. quantified spine density from the soma to distal tips in fixed brain sections and found a significant increase in spine density only at the distal tips of dendrites. In contrast, we quantified spine density using

in vivo imaging, and without limitation to distal tips[49]. Our results might therefore have missed local changes.

**Hypothetical models**. Because *Ndn* has pleiotropic functions as an adaptor protein or transcriptional cofactor and its alteration affects different neuronal cell types and at distinct developmental stages, the functional role of *Ndn* in Dup15q syndrome remains poorly understood. Considering our present and previous findings[9], however, we speculate on the possible mechanisms represented by three hypothetical models, namely *Ndn*–5-HT raphe; 5-HT-independent *Ndn* pathway; and altered SERT level in the developing cortex by *Ndn*.

For the *Ndn*–5-HT raphe pathway, previous studies revealed that *Ndn* is expressed in virtually all 5-HT neurons and that *Ndn* null mice show disturbed migration of 5-HT neuronal precursors, which in turn leads to altered global serotonergic architecture, increased spontaneous firing frequency and increased 5-HT reuptake via increased SERT expression[48,50–52]. In addition, we observed ameliorated resting membrane potential in the DRN of paternal *15q dupΔNdn* mice (Supplementary Fig. S9). Furthermore, our previous study showed chronic administration of fluoxetine, a selective serotonin reuptake inhibitor, could normalize the amplitude of sEPSC in the DRN of paternal *15q dup* mice[9]. Therefore, the synergistic effect of *Ndn* expression in DRN and 5-HT signaling may play a key role in the development of ASD-like behaviors or aberrant intrinsic spine dynamics found in paternal *15q dup* mice.

For 5-HT-independent *Ndn* pathway, we hypothesize here that the initial trigger is the decreased 5-HT seen in paternal *15q dup* mice. This in turn induces dysfunction at the synaptic or circuit level, given that 5-HT is required for normal cortical development and its alteration induces neurogenesis, cell migration, axon guidance, dendritogenesis, and synaptogenesis (reviewed in[53]). This alteration may be compensated by normalizing the expression of *Ndn* via a 5-HT-independent pathway in paternal *15q dupΔNdn* mice. This idea is supported by the results of the in utero electroporation experiment (Fig. 3). Overexpression of *Ndn* in cortical neurons induced the alteration of dendritic spines (i.e., without affecting raphe). Accordingly, 5-HT and *Ndn* may have independent pathways to dendritic spine formation, E/I balance and behaviors, but these might compensate for each other.

The third hypothesis is the possibility that SERT expression or activity in the developing cortex is altered in paternal *15q dup* and recovered in paternal *15q dupΔNdn* mice. While SERT is dominantly expressed in raphe neurons that synthesize 5-HT in the adult stage, strong and transient SERT expression is uniquely observed in a subset of glutamatergic neurons in several brain regions during development (E18 to around postnatal 2 weeks)[54]. The function of this cell type has not been fully elucidated; however, a very recent report revealed that SERT-expressing glutamatergic neurons in the prefrontal cortex (PFC) (named PFC-SERT+) regulate synaptic maturation of PFC-DRN circuits in a bidirectional manner[55]. Therefore, the primary target by *Ndn* may be in the PFC-SERT+ neurons, with a subsequent effect on the PFC-DRN circuit in the developmental period. In support of this hypothesis, early treatment (P3-P21) of SSRI to paternal *15q dup* mice alleviated several behavioral abnormalities and cortical E/I imbalance[9]. The increased expression of *Ndn* in paternal *15q dup* mice might accordingly produce a decrease in SERT expression in PFC-SERT neurons, followed by synaptic dysfunction in PFC-DRN.

For all hypotheses, it is important to assess (1) the crucial brain region or cell type affected by *Ndn*, (2) functional interaction partners or downstream molecules of *Ndn* for dendritic spine formation, and (3) relevance with SERT.

**Limitations**. Our study has some important limitations. First, this study was carried out in mouse models, and there seems to be different contributions from the two parental alleles to the phenotypes observed in humans and mice. Therefore, these results may not be directly relevant for explaining Dup15q etiology in humans. Matthew Anderson's group reported intriguing results[18]. They prepared two lines of BAC transgenic mice of *Ube3a*, a MEG. They named 1xTg and 2xTg, which had two- and threefold increased expression of *Ube3a* compared to WT mice, respectively. Therefore, the expression of *Ube3a* is postulated as equivalent levels between 1xTg and our maternal *15q dup* mice. Similar to our results, the 1xTg mice did not show impaired social behavior while the 2xTg mice exhibited ASD-like behavior. Furthermore, subjects having interstitial Dup15q (two maternal copies of *UBE3A*) often showed milder symptoms than isodicentric Dup15q (three maternal copies of *UBE3A*)[7]. From this evidence, mice may have higher resiliency for increased expression of *Ube3a* than that of humans, and the increased single copy number of *Ube3a* may not be sufficient to induce ASD-like phenotypes in mice. Compared to the MEGs, PEGs may have an opposite effect. In humans, paternal Dup15q is often normal (low penetrance and/or milder symptoms than that of maternal Dup15q), however, many abnormal phenotypes are found in mice from accumulated studies[8,9,13,14,23,56–58]. Thus, mice may have a higher susceptibility for the increased expression of PEGs than humans. We still do not know whether maternal and paternal Dup15q is an equivalent syndrome or not. Further investigation is required to clarify the difference between maternal and paternal Dup15q and the importance of paternal Dup15q.

In summary, our results identified *Ndn* as a driver gene for developing multiple ASD-like phenotypes found in paternal *15q dup* mice. Excess *Ndn* induced dysfunction in the cortical circuit, including dendritic spine formation/maturation and neuronal excitability in the cerebral cortex. Furthermore, normalizing the copy number of *Ndn* in paternal *15q dup* mice ameliorated abnormal dendritic spine turnover, cortical E/I imbalance and most ASD-like behaviors. These findings first suggested a PEG in 15q11-q13, *Ndn*, has the role of developing ASD-like abnormalities in mice.

## Methods

**Animals**. *15q dup* (also named *patDp/+*) mice were generated as previously described and maintained on C57BL/6J strain by backcrossing more than 10[8]. Paternal *Ndn* KO (*Ndn*[+m/−P]) mice were kindly provided by K. Yoshikawa (Osaka University). For visualizing dendritic spines, B6.Cg-Tg(Thy1-YFP)HJrs/J mice were used. The genetic backgrounds of these mice are maintained on C57BL/6J. *1.5 Mb Dp/+* and *15q dupΔNdn* mice were generated as described below and maintained on C57BL/6J. For in utero electroporation experiments, parous pregnant female mice were purchased (E15.5, C57BL/6J or ICR; SLC, Shizuoka, Japan). Mice were housed in a room with a 12-h light/dark cycle (light on 8:00 a.m. and off 8:00 p.m. in Hiroshima University and RIKEN; light on 6:00 a.m. and off 6:00 p.m. in Kobe University) and provided with ad libitum access to water and food. The breeding rooms were maintained at 22–28 °C with 50–60% humidity. All protocols for animal experiments were approved by Animal Care and Use Committees of Hiroshima University, RIKEN BSI, Nippon Medical School and Kobe University Graduate School of Medicine and performed under the institutional guidelines and regulations.

**Generation of 1.5 Mb duplication mice**. Mice with 1.5 Mb duplication were generated by in vivo *Cre*-mediated recombination *in trans* with *Hprt-Cre* transgenic mice according to previous reports[20,21,56]. Briefly, mice having a loxP sequence generated by random insertion of the transposon were provided by the TRACER database (SB-205515 and SB-197365)[59]. The genomic position of the integrated site was at 58,999,433 for SB-197365 and 60,474,596 for SB-205515 line, respectively (GRCm38/mm10). These mice were crossed with *Hprt-Cre* mice (maintained on C57BL/6J)[22]. From 105 animals screened by conventional genomic PCR, 2 *Dp/+* animals were generated. Primer sequences are available in Supplementary Table 1.

**Southern blot**. Genomic DNA was isolated from mouse tail by proteinase K/SDS digestion. Purified DNA (10 μg) was digested with BsrGI for *1.5 Mb Dp/+* mice and EcoRI for *15q dupΔNdn* mice, respectively. The digested DNA was electrophoresed on a 0.8% agarose gel and transferred to a Hybond-N + membrane (GE Healthcare UK Ltd., Buckinghamshire, UK). The membrane was hybridized with a digoxigenin-labeled DNA probe generated with a PCR DIG Probe Synthesis Kit (Merck KGaA, Darmstadt, Germany). For *1.5 Mb Dp/+* mice, the DNA probe was designed as a 560 bp fragment between the *Ube3a* and *Atp10a* genes, resulting in WT band: 6.6 Kb, *Dp* band: 14.8 Kb. For *15q dupΔNdn* mice, the DNA probe was designed as a 253 bp fragment 10 Kb upstream of the *Ndn* gene, resulting in WT band: 12.3 Kb, Dp band: 14.8 Kb. Next, the membrane was washed with low stringent buffer (2 × SSC, 0.1% SDS) and then with high stringent buffer (0.5 × SSC, 0.1% SDS) at 65 °C, then blocked with Blocking solution (Merck KGaA) and incubated with anti-DIG-AP Fab (1:10,000 dilution; cat. 11093274910, Merck KGaA). Finally, the immunoreactive signal was detected by the chemiluminescent detection system (Merck KGaA).

**Array comparative genomic hybridization**. aCGH was performed according to the manufacturer's protocol (SurePrint G3 Mouse CGH Microarray Kit, 1 ×1 M, #G4838A, Agilent Technologies, Inc., Santa Clara, CA, USA) and a previous study[56]. DNA from a male WT mouse (C57BL/6J) was used as reference genomic DNA (mm9). The extracted signals were then analyzed under the R environment with DNAcopy, Gviz, and GenomicRanges packages.

**Quantitative reverse transcription PCR**. Quantitative RT-PCR was performed as previously described with minor modifications[56]. Total RNAs in brain tissues or primary neurons were extracted with an SV total RNA isolation kit (Promega Corporation., Madison, WI, USA) or TRIZOL Reagent (Thermo Fisher Scientific) and reverse-transcribed with SuperScript II (Thermo Fisher Scientific, Waltham, MA, USA). *Gapdh* gene was used as an internal control. Primer sequences are shown in Supplementary Table 1.

**Behavioral analysis in *1.5 Mb patDp/+* mice**

*Generals*. All pups from a single litter remained with the mother until weaning or at least postnatal day 21 (P21). Then, 2–5 mice were housed with male littermates. In the behavioral analysis, only male mice were used. Mice were divided into two cohorts. In the first cohort ($N = 18$ for each genotype), the reciprocal social interaction test was performed, while the open field test and Barnes maze test were conducted in the second cohort ($N = 18$ for each genotype). All behavioral tests were conducted between 09.00 and 18.00 h during the light phase of the circadian cycle. All behavior tests were completed from age 8–17 weeks. Mice were habituated to the testing room for at least 30 min before starting the behavioral tests.

*Open field test*. The open field test was performed as described in a previous study with minor modification[13]. First, the subject mouse was placed in a corner of the open field apparatus (50 × 50 × 50 cm) illuminated at 100 lux. Locomotor activity and duration in the center area of the field (36%) were measured for 2 h. Data were collected and analyzed using Image with Time OFCR4 (O'hara, Tokyo, Japan). The distance in the center area:total distance ratio ("Anxiety index") in 30 min can be used as an index of anxiety-related responses[24,25].

*Reciprocal social interaction test*. Two mice with an identical genotype and sex housed in different cages (unfamiliar mice) were placed in the open field arena (50 × 50 × 50 cm) with dim light (10 lux) and allowed to move freely for 10 min. These two mice had an identical genotype and were also of the same age and sex, and similar in body weight. In this test, a pair of mice was used as a sample. Images were captured at the rate of two per second with a CCD camera located above the field and time engaged in social interaction was measured.

*Barnes maze test*. Spatial and reversal learning abilities were evaluated using a Barnes maze as described in a previously study[8]. The circular field (100 cm diameter) having 12 holes (4 cm diameter) was elevated 100 cm from the floor. The field was illuminated at over 1000 lux. A black Plexiglas escape box (17 × 13 × 7 cm) containing bedding material on its base was located under one of the holes. The hole above the escape box represented the target. The location of the target was consistent for a given mouse but randomized across mice. The maze was rotated before every trial, with the spatial location of the target unchanged with regard to distant visual room cues to prevent bias based on olfactory or proximal cues within the maze. Three trials per day were conducted on six successive days. On day 7, a probe trial was conducted without the escape box to confirm that this spatial task was acquired based on navigation by distal environment room cues. For the reversal task, the target location for each mouse was shifted to the complete opposite side on the circular surface. The same number of trials per day but for 4 days were also carried out and a probe trial was conducted. Time spent around each hole was recorded with automated tracking software (TimeBCM, O'hara).

**Behavioral analysis in *1.5 Mb matDp/+* mice**. All the *1.5 Mb matDp/+* and its control WT mice for behavior analyses were obtained by in vitro fertilization. After

weaning (3 weeks old), 2–5 mice were housed with same-sex littermates. In total, 17 WT (eight males and nine females) and 14 *1.5 Mb matDp/+* (seven males and seven females) were used for behavior tests. All behavioral tests were conducted between 09.00 and 18.00 h during the light phase of the circadian cycle. All behavior tests were completed from age 11 to 17 weeks. Mice were habituated to the testing room for at least 30 min before starting the behavioral tests. Behavioral order and method were the same as the analyses of *1.5 Mb patDp/+* mice described above.

**DNA constructs for in utero electroporation.** Plasmids of βActin promoter-driven EGFP (pβActin-EGFP) and its empty vector (pβActin-MCS) were gifted by S. Okabe (University of Tokyo). For overexpression of *Ndn*, the pRC-*Ndn* construct was gifted by K. Yoshikawa (Osaka University). For other genes, including *Snrpn, Magel2,* and *Mkrn3,* PCR product containing the coding region was subcloned into pβActin-MCS. The N-terminal deletion form of NDN (NDNΔN; AA110-325) was prepared by inverse PCR and subcloned into a pRC vector. DsRed was subcloned into pβActin-MCS (pβActin-DsRed). All plasmids were purified with a NucleoBond Xtra Midi kit (Takara Bio Inc., Shiga, Japan) and verified with DNA sequencing.

**In utero electroporation.** In utero electroporation was performed according to a previously described method with minor modification[12]. Briefly, E15.5 pregnant C57BL/6J or ICR mice (only for Supplementary Fig. 4) were anesthetized by inhalation of isoflurane (FUJIFILM Wako Pure Chemical Corporation, Osaka, Japan). The uterine horns were then exposed and 1–2 µl of plasmids were injected into one side of the lateral ventricle in the embryonic brain with a glass micropipette. Plasmids were diluted in HBS buffer (20% HEPES, 8% NaCl, 50 mM KCl, 7 mM $Na_2HPO_4$, and 1% glucose). For the overexpression of each 15q11-q13 gene, plasmids were mixed with pβActin-EGFP at a ratio of 1:1 (final concentration of plasmids: 0.5 µg/µl). For control, pβActin-EGFP and pβActin-DsRed were used in the same ratio (0.5 µg/µl each). A one-tenth amount of 0.1% Fast Green (Merck KGaA) solution was added to the plasmid mixture. Square electronic pulses (30 V for 50 ms, four times with 950 ms intervals) were delivered with a forceps-type electrode (CUY650P5; NEPAGENE, Chiba, Japan) connected to an electroporator (CUY21SC; NEPAGENE). The uterine horns were placed back into the abdominal cavity. After the operation, the body temperature of the mice was kept at 37 °C and the mice were returned to the home cage.

**Surgery and in vivo imaging.** Juvenile mice were used for in vivo imaging at postnatal 3 weeks. For in vivo two-photon imaging, thinned-skull methods were performed as described previously with minor modifications[12]. Mice were anesthetized with isoflurane inhalation during operation and imaging. The parietal bone was manually pruned using micro-surgical blades (Nordland Blade; Salvin Dental Specialities, Inc., Charlotte, NC, USA) until the thickness of the skull reached ~15 µm. After surgery, mice were set under the two-photon microscope system (A1R MP+; Nikon, Tokyo, Japan) with a pulsed laser (Chameleon Vision II; Coherent, Inc., Santa Clara, CA, USA) and ×25 water immersion objective lens (CFI75; Nikon, ApoLWD, numerical aperture (NA) 1.10). GFP (for plasmid overexpression) or YFP (for *Thy1-YFP-H*-line mice) fluorescence signal was separated with a dichroic mirror (560 nm) and barrier filter (575/25 nm). The target area was the somatosensory cortex, located at bregma—1.0 mm and lateral 1.5–2.5 mm., The wavelength was set to 920 nm to obtain fluorescence signals. To acquire dendritic spine images, the size of the image was 78 × 78 µm and the step size was set to 0.425 µm. The pixel size of single images was 512 × 512. The calculation method for the spine turnover rate was also as described previously[12]. After imaging on day 0, the scalp of mice was sutured and the mice were placed back into their home cage. On day 2, the scalp was re-opened and images of the same dendrite position were acquired. Images obtained on day 2 were used to calculate spine density. Some mice were excluded from spine turnover analysis (i.e., formation and elimination rate) and used only in density analysis because the spine turnover analysis required high-quality images.

**Classification of dendritic spines.** Dendritic spines were classified according to the previous studies[31]. Briefly, dendritic spines were commonly classified into four categories: Stubby, Filopodia, Thin, and Mushroom. "Stubby" spines had no neck and a small protrusion; "Filopodia" had no head and a long protrusion; "Thin" had a long thin neck with a small head; and "Mushroom" had a thin neck with a large head as a mature form. Spines were counted and classified into these four categories for control plasmid or *Ndn*-overexpressed neurons.

**Electrophysiology in cortical slices.** Mice with introduced *Ndn* or control with GFP plasmid by in utero electroporation were used in this analysis at 3 weeks old. Coronal slices (300 µm thickness) containing the somatosensory cortex barrel field (S1BF) were prepared with a VT1200S (Leica Microsystems, Wetzlar, Germany) in an ice-cold cutting solution (in mM): 120 choline-Cl, 2 KCl, 8 $MgCl_2$, 28 $NaHCO_3$, 1.25 $NaH_2PO_4$, and 20 glucose bubbled with 95% $O_2$ and 5% $CO_2$. The slices were incubated for at least 1 h in artificial cerebrospinal fluid (ACSF) (in mM): 125 NaCl, 2.5 KCl, 2 $CaCl_2$, 1 $MgSO_4$, 1.25 $NaH_2PO_4$, 26 $NaHCO_3$, and 20 glucose bubbled with 95% O2 and 5% CO2 at room temperature. The glass pipettes were

filled with an intracellular solution (in mM): 140 K-D-gluconate, 8 KCl, 2 NaCl, 10 HEPES, 0.2 EGTA, 3 Mg-ATP, and 0.5 $Na_2$-GTP adjusted to pH 7.3 with KOH. Whole-cell recordings were made from the somata of GFP-positive pyramidal neurons in layer II/III S1BF at 32 ± 1 °C. The GFP-positive pyramidal neurons were identified based on the triangular appearance of the cell soma and the presence of a single apical dendrite using an upright microscope (BX50WI; Olympus, Tokyo, Japan) with the emission of 470 nm LED light (SPECTRA; Lumencor, Beaverton, OR, USA). Action potentials were recorded at −70 mV holding potential with depolarized current injection. Miniature EPSCs were recorded at −80 mV holding potential in the presence of 0.5 µM tetrodotoxin and 10 µM bicuculline. Ionic currents were recorded with the EPC10 USB patch clamp amplifier (HEKA Elektronik, Lambrecht/Pfalz, Germany). Signals were digitized at 20 kHz for recording evoked responses or at 40 kHz for recording miniature responses. Online data acquisition and off-line data analysis were performed using Patchmaster software (V2x73.1, HEKA Elektronik). Miniature responses were analyzed using the Mini Analysis Program (ver. 6.0.7; Synaptosoft Inc., Decatur, GA, USA). Data analyses were performed in a blinded manner.

**Electrophysiology in the raphe nuclei.** Brain slices for experiments were prepared from 5-8 weeks old male mice described previously[9]. The mice were deeply anesthetized by isoflurane (~4% in air, v/v). Following decapitation, the brains were rapidly removed and placed in ice-cold $Na^+$-deficient saline (~4 °C) containing the following compounds: 252 mM sucrose, 21 mM $NaHCO_3$, 3.35 mM KCl, 0.5 mM $CaCl_2$, 6.0 mM $MgCl_2$, 0.6 mM $NaH_2PO_4$, and 10 mM glucose. Each brain was cut into blocks and three or four coronal slices (250 µm) were cut with a vibratome (VT1200s, Leica) through the entire rostro-caudal extent of the dorsal raphe nuclei (DRN) between −4.84 and −4.48 mm from the bregma according to the mouse brain atlas[60]. The slices were placed in a submerged chamber for at least 1 h in ACSF containing 138.6 mM NaCl, 3.35 mM KCl, 2 mM $CaCl_2$, 1.3 mM $MgCl_2$, 21.0 mM $NaHCO_3$, 0.6 mM $NaH_2PO_4$, and 10.0 mM glucose. The ACSF was maintained at pH 7.4 by bubbling with 95% $O_2$/5% $CO_2$ gas. Individual slices were transferred to a recording chamber attached to a microscope stage and continuously perfused with oxygenated ACSF at a flow rate of 1.4 ml/min, maintained at 29 °C. Borosilicate glass-patch electrodes (World Precision Instruments, Sarasota, FL, USA) were used for whole-cell recordings from DRN cells with resistance of ~3 MΩ when filled with an internal solution of 150 mM potassium methanesulfonate, 1.0 mM KCl, 0.2 mM K-EGTA, 20 mM HEPES, 3.0 mM Mg-ATP2, 0.4 mM Na-GTP, and 15 mM biocytin (pH 7.38, ~295 mOsm). DRN serotonergic cells were visually identified under IR-DIC images using a water immersion objective (×40, NA = 0.80; Olympus). Whole-cell patch clamp recordings were acquired and controlled using the Axon 700B Multiclamp amplifier (Molecular Devices, San Jose, CA, USA) and pClamp10 acquisition software (Molecular Devices, CA, US). The cell characteristics, including resting membrane potential and properties of the action potential, were measured in the current-clamp mode. For recording the spontaneous excitatory synaptic currents (sEPSCs), the cells were voltage-clamped at −70 mV. Serotonergic neurons were electrophysiologically identified by their action potential shapes having longer half-height width (>1.5 ms) and a large amplitude of overshoot (>30 mV), as previously reported[9]. All signals were filtered at 2 kHz and sampled at 10–20 kHz. Obtained data were analyzed by using software, clampfit11 (Molecular Devices), Kyplot (Kyence, Tokyo, Japan), and Mini analysis version 6 (Synaptosoft, Decatur, GA, USA).

**CRISPR/Cas9 construct and genomic PCR for evaluation of target deletion.** Target guide RNA sequence was determined with the CRISPR Design tool (http://crispr.mit.edu/). The 20-bp guide sequence for the proximal (sgNdnProx) and distal (sgNdnDist) region of *Ndn* were cloned into a pX330 vector (Addgene #42230). Target sequences were 5′-CCATGTAGATGACCGAAGTT-3′ for sgNdnProx and 5′-GTGCTAGAGATAAGATTCGT-3′ for sgNdnDist. A schematic chart is available in Supplementary Fig. 7a. According to the manufacturer's protocol, two guide RNA-expressing vectors (1 µg each) were transfected into the Neuro2a cell line by Lipofectoamine 3000 (Thermo Fisher Scientific). Genomic DNA was purified and target deletion was confirmed by conventional PCR. GFP plasmid was used as a negative control. Primer sequences are available in Supplementary Table 1.

**Generation of *15q dupΔNdn* mice.** Each of the sgRNA (sgNdnProx and sgNdnDist)-containing constructs was in vitro transcribed using a MEGA shortscript Kit (Thermo Fisher Scientific) and purified with a MEGA Clear Kit (Thermo Fisher Scientific). The target sgRNAs (50 ng/µl each) and *Cas9* mRNA (100 ng/µl, TriLink BioTechnologies, San Diego, CA, USA) were microinjected into the cytoplasm of fertilized eggs collected from *15q dup* mice (paternally derived duplication) as described in a previous study[61]. Injected embryos were re-implanted in ICR pseudo-pregnant females. Genotyping PCR was performed with deleted allele-specific primers, as shown in Supplementary Table 1.

**Digital PCR for evaluation of genomic copy number.** Genomic DNA for ddPCR was prepared from mouse tail using a DNeasy Blood & Tissue Kit (Qiagen, Hilden, Germany). ddPCR with the EvaGreen System was performed according to manufacturer's protocols (20 µl reactions as following components: 1x ddPCR SuperMix,

0.15 μM of each primer, 0.3 U/μl of EcoRI-HF (New England Biolabs, Ipswich, MA, USA), 1 ng/μl of genomic DNA). Droplets were then generated in the QX200 droplet generator (Bio-Rad Laboratories, Inc., Hercules, CA, USA). The plate was sealed with a PX1 PCR plate sealer (Bio-Rad). PCR reaction was performed using the following amplification protocol: 95 °C for 5 min, followed by 40 cycles: denaturation at 95 °C for 30 s; annealing and extension at 63 °C for 90 s, then incubation at 4 °C for 5 min followed by 90 °C for 5 min. Ramp speed was set to 2 °C/s. The amplicon was read using the QX200 droplet reader (Bio-Rad). *Htr1a* gene was used as an internal control. Primer sequences are available in Supplementary Table 1.

**Immunohistochemistry.** Immunohistochemistry for VGAT/VGLUT analysis was performed as previously reported with some modifications[9]. Mice aged 3 weeks old were anesthetized and transcardially perfused with phosphate buffer followed by 4% paraformaldehyde in 0.1 M PBS. Brains were dissected and immersed in the same fixative at 4 °C overnight. Coronal sections (50 μm thickness) were prepared with a vibratome (VT1200S, Leica), then stored in 0.1 M PBS containing 0.02% sodium azide at 4 °C. Brain sections were treated with 0.3% $H_2O_2$ in PBS for 30 min, washed with PBS for 5 min three times, incubated with the blocking solution (PBS containing 3% normal goat serum and 0.3% Triton X-100) for 30 min and incubated with primary antibodies including anti-VGAT rabbit polyclonal antibody (1:1000 dilution; cat. 131 003, Synaptic Systems, Goettingen, Germany), anti-VGLUT1 mouse monoclonal antibody (1:1000 dilution; cat. 135 311, Synaptic Systems), anti-GFP mouse monoclonal antibody (1:200 dilution; cat. A-11120, Thermo Fisher Scientific) or anti-Necdin rabbit antibody (1:200 dilution; NC243, Cosmo Bio Co., Ltd., Tokyo, Japan) at 4 °C overnight. Sections were washed four times with PBS containing 0.3% Triton X-100 (PBST) for 10 min each, incubated with fluorescent-conjugated secondary antibodies (Alexa Fluor 488 conjugated anti-mouse IgG, Alexa Fluor 488 conjugated anti-rabbit IgG, or Alexa Fluor 568 conjugated anti-rabbit IgG, Thermo Fisher Scientific) for 1 h, washed four times with PBST for 10 min each, transferred onto slide glass and mounted with VEC-TASHEILD with DAPI (VECTOR Laboratories, Inc., Burlingame, CA, USA). For VGAT/VGLUT analysis (Fig. 5f), images in cortical layer II/III of S1BF were obtained with a confocal laser scanning microscope FV3000 (Olympus) with a ×60 objective lens (digital zoom ×1.6). The pixel size of single images was 512 × 512. For quantification of the expression level of NDN (Supplementary Fig. 4), the same confocal microscope with a ×100 objective lens (digital zoom ×1.5) was used and Z-stacked images were obtained with 0.5 μm step (1024 × 1024 pixels). Then, the Z-projected images were obtained with maximum intensity. By seeing the GFP and DAPI signal, the transfected or non-transfected nuclei were manually selected ($N = 20$ for each). The mean intensity/pixel of NDN in the nucleus was used to compare transfected and non-transfected cells.

**VGAT/VGLUT puncta analysis.** To count VGAT and VGLUT puncta number, six images from three mice per each genotype were obtained in S1BF (Bregma, −1.34 to −1.82 mm). Then, three ROIs (100 × 100 pixels) from a single image were selected and converted to binary data with a predefined threshold. The threshold was determined by the linearity check. The mean puncta number of three ROIs was defined as a biological replicate. All analyses were performed by ImageJ software.

**Behavioral analysis in paternal *15q dupΔNdn* mice.** All behavior tests were conducted as described above (*1.5 Mb patDp/+* behavior section), except for the following.

*General.* Male mice were subjected to the following behavior tests in the order of open field test, reciprocal social interaction test, three-chambered social interaction test, female-induced male USV test, and Barnes maze test ($N = 18$ for WT, 19 for *15q dupΔNdn*). All behavior tests were completed at age 11–23 weeks except for the P7 USV test.

*Three-chambered social interaction test.* Sociability was evaluated by the three-chambered social interaction test[37]. Each chamber was 20 × 40 × 22 cm. Dividing walls were made of clear Plexiglass, with small square holes (5 × 3 cm) to allow access into each chamber (O'hara). An unfamiliar mouse (stranger; C57BL/6J) that had no prior contact with the subject mouse was placed in one of the chambers. The stranger mouse was enclosed in a small round wired cage. The subject mouse was first placed in the middle chamber and allowed to acclimate to the field for 10 min. Then, the sociability test was performed for 10 min (Stranger vs. Empty). Data were collected and analyzed using Time CSI (O'hara).

*Maternal isolation-induced pup USV.* The recording of USV emitted from pups was tested on postnatal day 7. Each pup was isolated from the dam and put into a plastic beaker containing bedding materials. Immediately, the beaker was placed in a sound-attenuated box and recording was started with an ultrasound detectable microphone (Avisoft-Recorder USGH, v3.4, Multi, 10311, Avisoft Bioacoustics, Berlin, Germany). The microphone was located 10 cm above the bottom of the field. The recording time was 5 min and the sampling rate was set to 300 kHz. Pups were transferred to a holding cage after recording. After all the pups in a cage had been recorded, they were transferred to their home cage. For acoustic analysis, the

data were transferred to SASLab Pro (Avisoft Bioacoustics) and fast Fourier transformed, with amplitudes under 30 kHz subsequently filtered out. The number of USVs was determined using SASLab Pro.

*Male USV during interaction with an estrus female.* USV emitted by a male mouse during interaction with female mice was measured as previously reported, with minor modifications[38]. Two days before testing, each subject male mouse was housed with an adult female mouse (C57BL/6J). The next day, male mice were transferred to a new cage and individually housed for 1 day. Early in the morning of the test day, the estrus cycle of female mice was determined. Each subject male mouse was transferred to the testing room and habituated for 30 min, followed by 10 min habituation to the testing field (cylinder; 23 cm diameter, 27 cm height). An estrus female mouse was therefore transferred to the testing field and USV was recorded for 5 min. The microphone was located 10 cm above the base of the field. The recording time was 5 min and the sampling rate was set to 300 kHz. For acoustical analysis, the data were transferred to SASLab Pro and fast Fourier transformed, and amplitudes under 30 kHz were subsequently filtered out. Because adult mice generate much noise that inhibits automatic analysis of USV, these noises were manually omitted and the call number was measured manually.

**Quantification of 5-HT and 5-HIAA in brain tissues.** Tissue contents of 5-HT and 5-HIAA in the brain were evaluated as described previously[13]. Briefly, each brain region (FC: frontal cortex, Hip: hippocampus, Hyp: hypothalamus, Mid: midbrain, CB: cerebellum, PoM: pons, and medulla) was dissected in ice-cold Hanks' balanced salt solution. Tissue homogenates were prepared by homogenization (0.2 M perchloric acid), deproteinization (on ice for 30 min) and centrifugation ($20,000 \times g$, for 15 min at 0 °C), filtration (0.45 μm; Merck KGaA) and pH adjustment to 3.0 with 1 M sodium acetate. The supernatant was used for high-performance liquid chromatography (HPLC) analysis (HTEC-500, Eicom, Kyoto, Japan). Tissue weight was used for the normalization of each sample.

**Serine content in the frontal cortex of *1.5 Mb Dp/+* mice.** The concentrations of L- and D-serine in brain tissues were measured by HPLC analysis. The frontal cerebral cortex was dissected and homogenized in 0.2 M ice-cold perchloric acid. The homogenates were prepared in the same way as above. L- and D-serine were then measured with an HPLC system (HTEC-500; Eicom). Precolumn derivatization was performed with 4 mM o-phthaldialdehyde/N-acetyl-L-cysteine (pH 10.0) in an auto sampling injector system (M-500; Eicom) at 10 °C for 5 min. The derivatives were separated in the octadecyl silane column (EX-3ODS, φ 4.6 mm × 100 mm long; Eicom) at 30 °C with 0.1 M phosphate buffer (pH 6.0) containing 18% methanol. The HPLC system was set to an applied potential of +600 mV vs. an Ag/AgCl reference analytical electrode. The amount of serine was determined with peak-area using PowerChrom (eDAQ, NSW, Australia) with external standards. Values were normalized by tissue weight.

**Primary hippocampal neurons of *1.5 Mb patDp/+* mice.** E16.5 pregnant mice of *1.5 Mb patDp+* were used to prepare primary neuronal culture. Embryos were surgically removed and hippocampi were dissected in ice-cold Hank's balanced salt solution. Then, primary hippocampal neurons were prepared by using SUMI-TOMO Nerve-Cell Culture System (Sumitomo Bakelite Co., Ltd). Prepared neurons were maintained in the neuronal plating medium containing MEM, 5% fetal bovine serum, 2 mM L-Glutamine, 0.6% D-glucose, 100 U/ml penicillin, and 100 μg/ml streptomycin at 37 °C, 5% $CO_2$ for 6 h. The medium was changed to neuronal maintenance medium (Neurobasal/NeuroBrew-21 supplement, 0.5 mM L-glutamine) until analysis. Neurons were plated on a poly-L-lysine-coated six-well plate at the density of $5 \times 10^5$ cells/well. On the day in vitro 7, neurons were washed with PBS and lysed in TRIZOL Reagent (Thermo Fisher Scientific) to extract total RNA.

**Statistics.** Statistical analysis was conducted using R and StatView ver5.0 (SAS Institute, Cary, NC, USA). Data were analyzed by two-way analysis of variance (ANOVA), two-way repeated measures ANOVA, or the *t*-test (two-tailed) unless otherwise noted. Specific statistical methods are described in the figure legend. Significance was set at $p < 0.05$. Hedges' g was calculated using the "effsize" package (version 0.7.4) in R[62].

**Reporting summary.** Further information on research design is available in the Nature Research Reporting Summary linked to this article.

## Data availability

There are no restrictions on data availability in this manuscript. The materials including *1.5 Mb Dp/+* or *15q dupΔNdn* mice are available from the corresponding author upon request. Mice having a loxP sequence generated by random insertion of the transposon were provided by the TRACER database Database (https://www.embl.de/distribution_spitz/distribution_spitz_1/) [TRACER database has been retired. Please contact mouse-informatics@ebi.ac.uk for more information]. Source data are provided with this paper.

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

## Acknowledgements

We thank K. Yoshikawa and K. Hasegawa for giving us the *Ndn* knockout mice and their valuable comments, H. Okubo for special technical assistance, and all the technical staff of the Takumi Lab for their technical assistance. This work was supported in part by KAKENHI (16H06316, 16H06463, 16K19066, 24700380, 20K07348, 21H00202, 21H04813) from the Japan Society for the Promotion of Science (JSPS) and the Ministry of Education, Culture, Sports, Science, and Technology; an Intramural Research Grant (30-9) for Neurological and Psychiatric Disorders of NCNP; the Takeda Science Foundation; Smoking Research Foundation; SENSHIN Medical Research Foundation; Tokyo Biochemical Research Foundation; Research Foundation for Opto-Science and Technology. K.F. was supported by JSPS fellowship (13J00993).

## Author contributions

K.T. and K.F. designed the experiments. K.T. performed RNA experiments, generated 1.5 Mb duplication and *15q dupΔNdn* mice, performed HPLC analysis and behavior experiments. K.F., T. Toya, S.T. and S.O. performed dendritic spine analysis. K.T. and J.A. performed E/I analysis. K.F and N.N. performed the electrophysiological analysis in the cerebral cortex. F. Saitow and H.S. performed the electrophysiological analysis in the raphe nuclei. F. Spitz provided mice having randomly integrated loxP sequence to generate micro-duplication mouse. K.T., K.F. and T. Takumi wrote the original manuscript. T. Takumi supervised the project.

## Competing interests

The authors declare no competing interests.
