## [Peer Review File · Nature Communications]

Reviewers' Comments:

Reviewer #1:

Remarks to the Author:

This manuscript describes a really nice series of studies that point to over-dosage of the imprinted gene *Necdin* being a key contributor to the phenotype associated with paternal 15q duplication. The authors have previously characterised a mouse model of paternal 15q duplication, and here they use a number of approaches to narrow down the key genes involved. Firstly, they generate a second 15q duplication mouse model (1.5Mb 15q dup) that has a smaller duplicated region and show that this model does not display many of the phenotypes seen in the original, larger 15q duplication model. This suggests that the genes (*Necdin*, *Mkrn3* and *Magel2*) outside this new duplication region are important. The authors then test this by using electroporation methods to over-express each gene individually and examining a dendritic spine morphology phenotype. These experiments point to *Necdin* being a major contributor, and the authors underline this by showing that knockout of *Necdin* in the original 15q duplication mouse model rescues many of the phenotypes.

The manuscript is excellent and the work described is novel and goes a long way to address the main question of the paper. I have a few issues that it would be good to address.

Firstly, the clarity of series of experiments could be improved. I think this relates partly to the naming of the two 15q duplication mouse models - the original being "15q dup" the new model being "1.5Mb 15q dup". When I first read the paper I was under the impression that the *Necdin* knockout rescue experiments were performed in the new model, which didn't make sense. It was only with careful re-reading that I realised that these 'rescue' experiments were performed in the original 15q dup mouse model. This needs to be made clearer somehow, probably with a few additional lines of text at the beginning of each Results section outlining what the experiments are more explicitly.

I was also curious as to why the authors did not show the expression levels of *Necdin* is altered following electroporation? How can they be sure that the over-expression of *Necdin* is equivalent to that of the other genes they screened (*Snrpn*, *Magel2* and *Mkrn3*)? After all this could explain the difference in degree of dendritic spine phenotype change between the different genes (Figure 3).

Minor issues:

Italicise gene names throughout

L. 47: I would suggest a change in wording. *Necdin* is not really an 'underestimated gene', as it has been extensively studied in the context of PWS (ie. loss of expression). I would suggest replacing with 'underestimated gene in the context of 15q duplication'

L. 86-88: This sentence implies there are a large number of individuals with Dup15 - in fact it is very rare. I think what the authors are suggesting is that the majority of those with 15q duplication have a maternally derived duplication. Please change to reflect this.

L. 90: Consider removing 'To the contrary' and beginning the sentence with 'However,'. The study cited was not contrary to the idea that maternal duplication is pathogenic (indeed this study still found that maternal 15q duplication was more penetrant), but as it was a much larger study it DID show that paternal duplications can be pathogenic for ASD (this is the point the authors are trying to make).

Reviewer #2:

Remarks to the Author:

This manuscript reports the finding that *Ndn* contributes to ASD phenotypes and neuronal morphology deficits in pat Dup15q mice. Although the authors did a lot of work, following a strong rationale, the conclusions are somewhat overstated and the relevance to human disease is incredibly inflated. Overall, I worry that this paper will muddy the waters in a field that desperately

needs to find the real genetic contributors to Dup15q syndrome, and this paper has little, if any, relevance to that, despite the strong statements and the title that would lead readers to think otherwise. My specific criticisms are detailed below.

Major issues:

1. The premise that the paternally expressed genes may contribute to the pathogenesis of the Dup15q syndrome is false. Dup15q syndrome is overwhelmingly caused by maternal duplications of 15q11-q13. Some individuals with maternal duplications inherited their CNV from an overtly neurotypical mother with the same CNV on her paternal allele. Individuals with paternal 15q11-q13 duplications while perhaps not fully neurotypical have a different disorder than Dup15q syndrome (see Cook et al., 1997). Even in the Isles et al., 2016 paper referenced as evidence for paternal duplications being involved in ASD, out of 10 individuals with paternally-inherited duplications, none have ASD (see suppl. Table 1).
2. The behaviors etc... of the maternal inheritance of the 1.5 Mb duplication are not discussed here. This is important because dogma (based on human genetics) suggests that Ube3a, which has maternal-only expression in neurons, is the most important gene for Dup15q syndrome. This hypothesis was never tested and proven or disproven.
3. The data from the Ndn knockout mice shows a very subtle change in dendritic spine formation rate. Furthermore, the induced neurons studies in Supplementary fig 3 are not clearly described. Are these from knockout animals or are they transfected constructs?
4. Overexpression studies never quantify the amount of overexpression of Ndn per neuron. Transfection often leads to grossly overexpressed gene products and this is not the same as an additional single genetic copy with its native regulatory elements.
5. The analysis of the Ndn knockout dup mice did not consider whether the DNA methylation of the upstream 15q genes was disrupted. There are DMRs in Ndn and Mkrn3 that impact the expression of genes in the region. Similarly, the parent of origin of the Ndn deletion was not determined. Nor was the gene expression of the genes in the dup-delta-Ndn compared to the dup. The expression of Mkrn3 is not consistent with copy number. The authors should use the PCR assay used to verify CRISPR cutting in Neuro2A cells in their mouse, since the Southern data is not terribly clear.
6. The statement on page 13, line 284-285 "These data indicated Ndn has a predominant role in the proper development of cortical circuits among 15q11-q13 genes" is not true. These data really looked at the genes on mouse chromosome 7 and are only relevant to the mouse. The statement "These data indicated Ndn has a predominant role in the proper development of cortical circuits among mouse chromosome 7C imprinted genes" is more appropriate.
7. The behaviors of the dup-delta-Ndn weren't compared to the dup mice. The behaviors in this mouse model are relatively subtle, and mouse behaviors can change over time and in different hands due to environmental and other influences. These mice should be directly compared.
8. The statement on page 14, line 320 that "Ndn is a new risk gene for ASDs" is wildly overstated. Human data suggest otherwise. The author points to one paper in which two individuals have a paternal duplication encompassing NDN, MAGEL2, and MKRN3, however, the Gnomad SV data report 8 alleles (individuals) with duplications of NDN, MAGEL2, and MKRN3 in neurotypical individuals, 5 of which are in the non-neuro cohort.
9. Overall, these data may be true specifically for this mouse model, which never really provided good face validity for human Dup15q syndrome. Therefore, the findings are only really relevant for mouse neurobiology. The language in the introduction and conclusions should really be tempered to reflect the limited applicability of this mouse model to the human condition.

Reviewer #3:

Remarks to the Author:

In this manuscript Tamada et al report new findings from analysis of a new line of mutant mouse model homologues to smaller region from AS imprinting center and UBE3A within the human 15q11-q13 region and other related studies. They conclude that the paternally expressed Ndn is a driver gene for the neurodevelopmental defect reported in the human paternal 15q11-q13 duplication syndrome.

Maternal duplication of human chromosome 15q11-q13 region (Dup15q) has been strongly implicated in autism spectrum disorder (ASD). While the pathogenic role of paternally derived duplication remains uncertain. A recent genetic study has implicated a significant contribution of paternal duplication to ASD/DD/MCA with penetrance rate of ~20%. While the total number of subjects analyzed is large in this report but the number of paternal Dup15q case is still relatively small (~10). So the conclusion regarding the role of paternal Dup15q remains somewhat questionable but is stronger than previous data in literature.

The same group has reported the mutant mice carrying the large 6Mb duplication that mimic the common human 15q11-q13 duplication before. Unexpected, the paternal but not maternal duplication display some autism-like behaviors. It remains unclear why maternal duplication does not contribute any significant defect in mice. In this study, the author has produced a new line of smaller duplication (upstream of AS-IC and downstream of Ube3a within the homologous region of 15q11-q13). The paternal duplication of this smaller region did not result in the similar behavioral or cellular phenotypes of that observed in the large 6Mb duplication. It is noted that the same analysis of the small maternal duplication was not included in manuscript and could be valuable. Because of these results, the focus of the study was then turned to the paternally expressed genes proximal to the small duplicated region. A series of experiments including in vivo rescue has been conducted. The results provided the support that Ndn is the critical gene to responsible for the autism phenotypes of the large 6 Mb paternal duplication in mice. Overall, authors should be complimented on the large number of experiments that are both in vitro and in vivo and at cellular, functional and behavioral levels. The conclusion is reasonably convincing based on the results presented. However, several issues may be worthwhile to discuss in revision

Major concerns:

1. The reference #17 is cited as the major evidence supporting the pathogenicity of the paternal duplication of 15q11-q13. The large sample size was repeatedly mentioned. This statement may be misleading. The large population was screened for the 15q11-q13 CNV but the total number of cases for genotype and phenotype study of the paternal duplication is ~10 and small. In addition, the penetrance of paternal duplication is ~20% which indicate that many cases carrying the paternal duplications are not affected. This caveats of this report should be discussed in detail.
2. In figure 1 D, the level of the increased expression is Ube3a is less than other paternally expressed genes but it is statistically significant based on the data presented. In addition, because the maternal allelic expression of Ube3a is cell type specific, the total RNAs from the brain tissue with mixed cell types may underestimate the difference too.
3. While the results from the 15q dup Δ Ndn mice provide reasonable evidence; because both behaviors and spine morphology are less specific as the readout for the manipulation, it seem that the more important rescuing experiments may be the neurochemical analysis of quantification of 5-HT and 5-HIAA in brain if technically feasible and at least electrophysiological recording experiment what should be feasible. Both of these were performed in 15q dup Δ Ndn mice and the large 6Mb duplication mice
4. A similar restriction enzyme map that illustrates the genomic Southern analysis of 1.5 Mb duplication mice described in Figure 1 may help reader and future reference
5. Two different social behavior paradigms were used in this and previous paper. Only 3 chamber sociability test was used in the 6Mb paternal duplication mice.. Both reciprocal social interaction test and 3 chamber sociability test were used in mice with 15q dup Δ Ndn. The inconsistency among

the experimental design may confound the data interpretation and should be discussed.

6. The interpretation and statement of E/I imbalance in text may be a bit premature because only the function of pyramidal neurons was measured and the E/I imbalance should be a net effect of E/I either at single cell or network level.

7. Line 361: "We are the first to identify Ndn as a causative gene for Dup15q syndrome". This may be an overstatement. Not sure the author has enough to conclude the NDN is the "causative" gene for human Dup15q syndrome based on the animal study. The conclusion for a causative role of a gene implicated in human should only come from human data. The animal and in vitro data may provide some evidence to support the case. Unless the phenotype is pathognomonic for human diseases, these evidence is not direct.

Minor concerns

1. There are many discussion and comments in result sections. In general, these comments should be reserved in discussion section.

2. Line 61: Autism spectrum disorder (ASD) is a neurodevelopmental disorder. ASD is not "a" disorder

3. Line 61-62: The prevalence rate of ASD is now estimated 1 in 59 according to the Centers for Disease Control (CDC). Add the reference for this statement

4. Line 64-65: "including rare variants of a single gene or pathogenic copy number variations (CNVs)": rare variants of a single gene are also pathogenic

5. Line 66-68: There are reported association between CNVs and schizophrenia, but few evidence supporting the causality such as CNV in 15q11-q13. So be cautious to make a generalized statement

6. One of the interesting and also intriguing questions remains why the maternal duplication of the region in the mice has largely not be able to recapitulate the human maternal duplication.

We thank the reviewers for the careful review of our manuscript and for many useful suggestions to improve it. Our responses to the points raised are as follows:

<Reviewer #1>

*This manuscript describes a really nice series of studies that point to over-dosage of the imprinted gene *Necdin* being a key contributor to the phenotype associated with paternal 15q duplication. The authors have previously characterised a mouse model of paternal 15q duplication, and here they use a number of approaches to narrow down the key genes involved. Firstly, they generate a second 15q duplication mouse model (1.5Mb 15q dup) that has a smaller duplicated region and show that this model does not display many of the phenotypes seen in the original, larger 15q duplication model. This suggests that the genes (*Necdin*, *Mkx3* and *Magel2*) outside this new duplication region are important. The authors then test this by using electroporation methods to over-express each gene individually and examining a dendritic spine morphology phenotype. These experiments point to *Necdin* being a major contributor, and the authors underline this by showing that knockout of *Necdin* in the original 15q duplication mouse model rescues many of the phenotypes.*

The manuscript is excellent and the work described is novel and goes a long way to address the main question of the paper. I have a few issues that it would be good to address.

*Firstly, the clarity of series of experiments could be improved. I think this relates partly to the naming of the two 15q duplication mouse models - the original being "15q dup" the new model being "1.5Mb 15q dup". When I first read the paper I was under the impression that the *Necdin* knockout rescue experiments were performed in the new model, which didn't make sense. It was only with careful re-reading that I realised that these 'rescue' experiments were performed in the original 15q dup mouse model. This needs to be made clearer somehow, probably with a few additional lines of text at the beginning of each Results section outlining what the experiments are more explicitly.*

We appreciate the reviewer's positive comments. We understand the confusion of the naming of a mouse. We have made the new Supplementary Fig. 1, which

indicates details of the duplicated region with 1.5 Mb, and we hope this new figure helps the reader's understanding. We have also included additional explanations to be made clearer in the revised manuscript. For *15q dupΔNdn* mice, we modified the sentence to be clearer as follows:

Line 246 (revised manuscript)

“Therefore, we examined whether the normalization of the genomic copy number of *Ndn* in *15q dup* mice is sufficient to restore the abnormal phenotypes”

=>

“Therefore, we examined whether removing a single copy of *Ndn* from original *15q dup* mice is sufficient to restore the abnormal phenotypes”

*I was also curious as to why the authors did not show the expression levels of *Necdin* is altered following electroporation? How can they be sure that the over-expression of *Necdin* is equivalent to that of the other genes they screened (*Snrpn*, *Magel2* and *Mkrn3*)? After all this could explain the difference in degree of dendritic spine phenotype change between the different genes (Figure 3).*

According to the reviewer's comments, we quantified the expression level of *Ndn* both in transfected and non-transfected cells induced by *in utero* electroporation followed by immunohistochemistry (new Supplementary Fig. 4). We observed an about 3-fold increased level in transfected cells than non-transfected cells. Since the reliable antibodies for MAGEL2, SNRPN, or MKRN3 were not available, we could not quantify them accurately. However, we thought the expression levels of these genes are similar because the other plasmids have the same promoter and vector. Based on this result, we corrected the manuscript as follows:

Result

Line 183 (revised manuscript)

“The overexpression level of *Ndn* was verified by immunohistochemistry, and it showed about 3-fold increase compared to non-transfected neurons (Supplementary Fig. 4).”

Material and Methods

Line 658 (revised manuscript)

“Briefly, E15.5 pregnant C57BL/6J mice were anesthetized by inhalation of isoflurane”

=>

“Briefly, E15.5 pregnant C57BL/6J mice or ICR (only for Supplementary Fig. 4) were anesthetized by inhalation of isoflurane”

Line 819 (revised manuscript)

“For VGAT/VGLUT analysis (Fig. 5f),”

Line 822 (revised manuscript)

“For quantification of the NDN expression level (Supplementary Fig. 4), the same confocal microscope with a 100x objective lens (digital zoom x 1.5) were used, and Z-stacked images were obtained with 0.5 μm step (1024 x 1024 pixels). Then, the Z-projected images were obtained with maximum intensity. By seeing the GFP and DAPI signal, the transfected or non-transfected nuclei were manually selected (N = 20 for each). Mean intensity/pixel of NDN in the nucleus was used to compare transfected and non-transfected cells.”

Minor issues:

Italicise gene names throughout

We corrected them throughout the manuscript.

L. 47: I would suggest a change in wording. Necdin is not really an 'underestimated gene', as it has been extensively studied in the context of PWS (ie. loss of expression). I would suggest replacing with 'underestimated gene in the context of 15q duplication'

As the reviewer suggested, we corrected it.

Line 49 (revised manuscript)

L. 86-88: This sentence implies there are a large number of individuals with Dup15 - in

fact it is very rare. I think what the authors are suggesting is that the majority of those with 15q duplication have a maternally derived duplication. Please change to reflect this.

According to the reviewer's comment, we corrected as below:

Line 44 (revised manuscript)

“However, a recent genetic study with a large sample size of Dup15q has implicated a significant...”

=>

“However, a recent genetic study has implicated a significant...”

Line 87 (revised manuscript)

"The large number of..."

=>

"Most"

L. 90: Consider removing 'To the contrary' and beginning the sentence with 'However;'. The study cited was not contrary to the idea that maternal duplication is pathogenic (indeed this study still found that maternal 15q duplication was more penetrant), but as it was a much larger study it DID show that paternal duplications can be pathogenic for ASD (this is the point the authors are trying to make).

We agree with the reviewer's comment. We corrected as below:

Line 91 (revised manuscript)

"To the contrary, however, an investigation with a large sample size found individuals with paternally derived duplication also met the criteria for ASD."

=>

" In contrast, an investigation found individuals with paternally derived duplication also met the criteria for ASD, although its penetrance was estimated at ~20% and the number of cases is still small¹⁹."

<Reviewer #2>

This manuscript reports the finding that Ndn contributes to ASD phenotypes and neuronal morphology deficits in pat Dup15q mice. Although the authors did a lot of work ,

following a strong rationale, the conclusions are somewhat overstated and the relevance to human disease is incredibly inflated. Overall, I worry that this paper will muddy the waters in a field that desperately needs to find the real genetic contributors to Dup15q syndrome, and this paper has little, if any, relevance to that, despite the strong statements and the title that would lead readers to think otherwise. My specific criticisms are detailed below.

We appreciate the reviewer's important suggestions. Although, in our original manuscript, we emphasized that *Ube3a* is a primary gene for Dup15q considering from human genetics studies, we have edited the whole manuscript, including the title, and also added "Limitations" in the Discussion section (revised manuscript line: 468-488) to avoid overstatement of our conclusion in our revised manuscript.

Title

“Genetic dissection identifies Necdin as a driver gene in 15q duplication syndrome”

=>

“Genetic dissection identifies Necdin as a driver gene in a mouse model of 15q duplication syndrome”

Major issues:

1. The premise that the paternally expressed genes may contribute to the pathogenesis of the Dup15q syndrome is false. Dup15q syndrome is overwhelmingly caused by maternal duplications of 15q11-q13. Some individuals with maternal duplications inherited their CNV from an overtly neurotypical mother with the same CNV on her paternal allele. Individuals with paternal 15q11-q13 duplications while perhaps not fully neurotypical have a different disorder than Dup15q syndrome (see Cook et al., 1997). Even in the Isles et al., 2016 paper referenced as evidence for paternal duplications being involved in ASD, out of 10 individuals with paternally-inherited duplications, none have ASD (see suppl. Table 1).

We again agree that Dup15q syndrome is predominantly caused by maternal duplication. Indeed, we emphasized a maternal duplication is a dominant form in

Dup15q syndrome repeatedly throughout our original manuscript. According to the reviewer's comment, we overall corrected the expression of the manuscript to be more precise and add a "Limitation" section (revised manuscript line: 468-488). In the article of Isles et al., we found 2 individuals with ASD and 1 "Autistic traits" out of 10 paternal duplications (suppl. Table 1, the modified table is shown below).

		Patient source			15q11-q13 duplication details						Neuropsychiatric features			
No	ID	Publication and/or source	Ascertainment	Start	Stop	Size (Mb)	Breakpoints	Genome build	Inheritance of duplication	Parental Origin	Developmental delay/Intellectual disability	ASD	Behavioural	Other
10	2825	Gawlik M	Transmitting parent	22652330	28535267	5.88	BP1-BP3	hg 19	Unknown	Paternal				
16	Bonn_Munich1	Priebe et al., 2013	SZ	22416000	28488000	6.07	BP1-BP3	hg 19	Unknown	Paternal	WAIS: FSIQ 72. Hauptschule (regular graduation after 9 years), no further schooling, slight learning disability		Shy and jumpy as a child, difficulty establishing contact; thumb-sucking after age 5y; Poly substance abuse; Sleep disturbances in childhood; Psychomotor agitation; Disorganized behaviour	
19	NF309	Aleksic B, Itaru K	Transmitting parent	22416000	28488000	6.07	BP1-BP3	hg 19	Unknown	Paternal				
27	Control 1	Ingason et al., 2011	Control	21240037	26208861	4.96	BP2-BP3	hg 18	Paternal	Paternal				
28	Father of Control 1	Ingason et al., 2011	Father of control 1	21240037	26208861	4.96	BP2-BP3	hg 18	De novo	Paternal				
29	Control 2	Ingason et al., 2011	Control	20306549	26208861	5.9	BP1-BP3	hg 18	Unknown	Paternal	IQ=100, GAF=81, dyslexia			
35	Ingason, new carrier 5	Ingason et al., 2011	Control	21240037	26208861	4.96	BP2-BP3	hg 18	De novo	Paternal				
44	S0320-3	Stergakouli et al 2012	ADHD	23683783	28501101	4.81	BP2-BP3	hg 19	Paternal	Paternal	FSIQ 70	Autistic traits		
46	J. Oglvie	Ahn et al. 2013	ASD, DD/ID	23656936	28526440	4.86	BP2-BP3	hg19	De novo	Paternal	Developmental delay (severity unspecified)	ASD		
47	L. Oglvie	Ahn et al. 2013	ASD	22765627	28940098	6.16	BP1-BP3	hg19	De novo	Paternal		ASD		

2. The behaviors etc... of the maternal inheritance of the 1.5 Mb duplication are not discussed here. This is important because dogma (based on human genetics) suggests that *Ube3a*, which has maternal-only expression in neurons, is the most important gene for Dup15q syndrome. This hypothesis was never tested and proven or disproven.

Thank you for the reviewer's suggestion. Our hypothesis was based on mouse data. Maternal 15q dup mice (i.e., 6.3 Mb) and *Ube3a* BAC transgenic (1xTg) mice did not show autistic-like abnormalities (Nakatani et al., 2009; Smith et al., 2010). We have described this issue in a "Limitations" section (revised manuscript line: 468-488). In addition, we re-considered the reviewer's concern and conducted the behavioral examination of 1.5 Mb *matDp*/+ mice (Supplementary Fig. 3). 1.5 Mb *matDp*/+ mice did not exhibit autistic-like behaviors like the original 6.3 Mb *matDp*/+ mice.

3. The data from the *Ndn* knockout mice shows a very subtle change in dendritic spine formation rate. Furthermore, the induced neurons studies in Supplementary fig 3 are not clearly described. Are these from knockout animals or are they transfected constructs?

As the reviewer pointed out, the change in the spine formation rate of *Ndn* KO mice was smaller than that of overexpression of *Ndn*. The gain of function of a gene affects the phenotype more than the loss of function. This is supported by accumulated animal studies, such as CAG-repeat gene or Alzheimer-related gene, that transgenic animals had severe phenotypes than KO animals. Therefore, the subtle change in KO of the gene is not so special, which does not affect our conclusion. Regarding Supplementary Fig. 3 (new Supplementary Fig. 6), these data were obtained with transfected constructs. To make it clear, we corrected the Figure legend and a Result section as follows:

Result

Line 213 (revised manuscript)

We added “construct” and “ with overexpression of a deletion construct ”

Figure legend

Line 1233 (revised manuscript)

We added, "each construct of".

4. Overexpression studies never quantify the amount of overexpression of Ndn per neuron. Transfection often leads to grossly overexpressed gene products and this is not the same as an additional single genetic copy with its native regulatory elements.

Thank you for this suggestion. We quantified the expression level of *Ndn* induced by in utero electroporation (new Supplementary Fig. 4). As responded to reviewer 1, we observed an about 3-fold increased level in transfected cells than non-transfected cells. Since the reliable antibodies for MAGEL2, SNRPN, or MKRN3 were not available, we could not quantify them accurately. However, we thought the expression levels of these genes are similar because the other plasmids have the same promoter and vector. Regarding the reviewer’s concern ("*Transfection often leads to grossly overexpressed gene products and this is not the same as an additional single genetic copy with its native regulatory elements.*"), this is why we prepared and analyzed *15q dupΔNdn* mice. We had already described in the original manuscript as follows:

Line 245 (revised manuscript) and line 242 (original manuscript)

" It is possible that the overexpression of Ndn achieved using in utero electroporation with a strong artificial promoter may not reflect physiological conditions. Therefore, we examined whether the normalization of the genomic copy number of Ndn in 15q dup mice is sufficient to restore the abnormal phenotypes"

5. The analysis of the *Ndn* knockout *dup* mice did not consider whether the DNA methylation of the upstream 15q genes was disrupted. There are DMRs in *Ndn* and *Mkrn3* that impact the expression of genes in the region. Similarly, the parent of origin of the *Ndn* deletion was not determined. Nor was the gene expression of the genes in the *dup-delta-Ndn* compared to the *dup*. The expression of *Mkrn3* is not consistent with copy number. The authors should use the PCR assay used to verify CRISPR cutting in *Neuro2A* cells in their mouse, since the Southern data is not terribly clear.

Thank you for these comments. Although we did not examine the methylation level of *Ndn* or 15q genes in *15q dupΔNdn* mice, we quantified mRNA levels of *Ndn* and its neighboring genes in original Figure 5c and found only *Ndn* expression was normalized to the WT level. The origin of *Ndn* deletion in *15q dupΔNdn* mice was paternally derived duplication. We have added the information in a Material and Method section of our revised manuscript (line 782). In addition, in revised Figure 5c, we directly compared the expression of all genes in the duplicated region between *15q dup* and *15q dupΔNdn* mice. As expected, only *Ndn* expression was normalized to the WT level, whereas other 15q11-q13 genes in *15q dupΔNdn* mice still showed a similar level as *15q dup* mice. As the reviewer suggested, the expression level of *Mkrn3* is higher than the expected level from the copy number. The increased *Mkrn3* expression was also seen in *15q dup* and *15q dupΔNdn* mice (new Fig. 5c). This means that the increased expression of *Mkrn3* is not due to the CRISPR-Cas9 mediated *Ndn* deletion. Currently, we do not have any direct evidence for explaining the reason for anomaly increased expression of *Mkrn3*; however, the absolute expression level of *Mkrn3* is quite low in the adult brain as reported (Liu et al., *Oncotarget*, 8, 85102-85109, 2017). It might reflect the floor effect. We had already validated CRISPR/Cas9 efficiency in *Neuro2a* cells (original

Supplementary Fig. 4a and b). According to the reviewer's comment, we have included the data of CRISPR/Cas9 mediated genomic deletion in mice (new Supplementary Fig. 7e) in this revised manuscript.

6. *The statement on page 13, line 284-285 "These data indicated *Ndn* has a predominant role in the proper development of cortical circuits among 15q11-q13 genes" is not true. These data really looked at the genes on mouse chromosome 7 and are only relevant to the mouse. The statement "These data indicated *Ndn* has a predominant role in the proper development of cortical circuits among mouse chromosome 7C imprinted genes" is more appropriate.*

We agree with the reviewer's comment and have corrected the expression of the revised manuscript as the reviewer suggested.

Line 287 (revised manuscript)

"These data indicated *Ndn* has a predominant role in the proper development of cortical circuits among 15q11-q13 genes"

=>

"These data indicated *Ndn* has a predominant role in the proper development of cortical circuits among mouse homologous genes in 15q11-q13."

7. *The behaviors of the dup-delta-*Ndn* weren't compared to the dup mice. The behaviors in this mouse model are relatively subtle, and mouse behaviors can change over time and in different hands due to environmental and other influences. These mice should be directly compared.*

As the reviewer suggested, we agree that ideally, it is better to compare their behavior among 4 groups (littermate WT of *15q dup*, *15q dup*, littermate WT of *15q dupΔNdn*, and *15q dupΔNdn*). However, not only a battery of behavioral tests using these 4 groups all together are physically challenging but also we have already reported the behavioral abnormalities of *15q dup* mice multiple times in addition to the original study published in 2009, and those behavioral tests were independently performed in different institutes (Tamada et al., 2010; Nakai et al., 2017; Nagano et al., 2018; Tsurugizawa et al., 2020). Furthermore, we used the control littermate

animals (i.e., WT) in every analysis to cancel the environmental effect as the reviewer suggested.

8. The statement on page 14, line 320 that “Ndn is a new risk gene for ASDs” is wildly overstated. Human data suggest otherwise. The author points to one paper in which two individuals have a paternal duplication encompassing NDN, MAGEL2, and MKRN3, however, the Gnomad SV data report 8 alleles (individuals) with duplications of NDN, MAGEL2, and MKRN3 in neurotypical individuals, 5 of which are in the non-neuro cohort.

According to the reviewer’s suggestion, we have corrected the summary section and added a "Limitations" section in Discussion to emphasize that this study is conducted in not human but mice. We have also described the low penetrance of paternal Dup15q in a “Limitations” section (revised manuscript line: 468-488).

Line 335 (revised manuscript)

"Ndn, is a new risk gene for ASDs."

=>

"Ndn, has a critical role for developing ASD-related phenotypes seen in 15q dup mice."

Line 482 (revised manuscript)

“In human, paternal Dup15q is often normal (low penetrance) ...”

9. Overall, these data may be true specifically for this mouse model, which never really provided good face validity for human Dup15q syndrome. Therefore, the findings are only really relevant for mouse neurobiology. The language in the introduction and conclusions should really be tempered to reflect the limited applicability of this mouse model to the human condition.

Again, according to the reviewer’s suggestion, we have corrected the indicated points throughout the whole manuscript and added a "Limitations" section (revised manuscript line: 468-488) which emphasizes that this study is conducted in mice

and maternal duplication is strongly implicated in Dup15q syndrome.

Title

“Genetic dissection identifies *Necdin* as a driver gene in 15q duplication syndrome”

=>

“Genetic dissection identifies *Necdin* as a driver gene in 15q duplication syndrome in a mouse model”

Abstract

Line 51 (revised manuscript)

We added “in mice” at the end of the sentence.

Line 57 (revised manuscript)

“Taken together, this study provides the significant contribution of the paternally expressed gene in 15q11-q13 for developing ASD and *Ndn* is a potential candidate for further investigation in neurodevelopmental disorders.”

=>

“Taken together, this study provides the significant contribution of *Ndn* in 15q dup mice and *Ndn* is a potential candidate for further investigation in neurodevelopmental disorders.”

Introduction

Line 91 (revised manuscript)

"To the contrary, however, an investigation with a large sample size found individuals with paternally derived duplication also met the criteria for ASD."

=>

" In contrast, an investigation found individuals with paternally derived duplication also met the criteria for ASD, although its penetrance was estimated at ~20% and the number of cases is still small"

Line 93 (revised manuscript)

“Thus, it remains unclear whether paternally expressed genes (PEGs) in 15q11-q13 contribute to the pathogenesis of Dup15q syndrome.”

=>

“Thus, it remains unclear whether paternally expressed genes (PEGs) in 15q11-q13 contribute to the pathogenesis of ASD.”

Line 95 (revised manuscript)

“To address the contribution of PEGs in 15q11-q13 for Dup15q syndrome,”

=>

“To address the contribution of PEGs in 15q11-q13 for ASD,”

Discussion

Line 335 (revised manuscript)

"Ndn, is a new risk gene for ASDs."

=>

"Ndn, has a critical role for developing ASD-related phenotypes seen in 15q dup mice."

Line 343 (revised manuscript)

“Due to the large number of individuals with its maternally derived duplication,”

=>

“Due to the predominant number of individuals with its maternally derived duplication,”

Line 351 (revised manuscript)

“Here, therefore, we tried ...using a Dup15q model mouse.”

=>

“Therefore, we tried ...using a model mouse.”

Line 338 (original manuscript)

“This 15q dup mouse is a kind of artificially generated founder mouse for forward genetics of Dup15q syndrome”

=> removed

Line 353 (revised manuscript)

“We are the first to identify *Ndn* as a causative gene for Dup15q syndrome”

=>

“We are the first to identify *Ndn* as a critical gene in 15q11-q13 for developing of autistic-like behaviors in mice”

Line 381 (revised manuscript)

“...reported that paternal duplication leads to an increased risk...”

=>

“...reported that paternal duplication might lead to an increased risk...”

Line 384 (revised manuscript)

“These studies indicate the importance of PEG in the development of Dup15q syndrome, in addition to *UBE3A*.”

=>

“These studies may suggest the relevance of PEG and Dup15q syndrome in addition to *UBE3A* though paternal and maternal Dup15q may be different syndrome.

Line 491 (revised manuscript)

“In summary, our results identified *Ndn* as a driver gene for Dup15q syndrome”

=>

“In summary, our results identified *Ndn* as a driver gene for developing multiple autistic-like phenotypes found in *15q dup* mice.”

Line 496 (revised manuscript)

“These findings have revealed the role of PEGs in 15q11-q13 in the pathogenesis of ASD”

=>

“These findings firstly suggested a PEGs in 15q11-q13, *Ndn*, has the role for developing autism-related abnormalities in mice.”

<Reviewer #3>

*In this manuscript Tamada et al report new findings from analysis of a new line of mutant mouse model homologues to smaller region from AS imprinting center and *UBE3A* within*

the human 15q11-q13 region and other related studies. They conclude that the paternally expressed Ndn is a driver gene for the neurodevelopmental defect reported in the human paternal 15q11-q13 duplication syndrome.

Maternal duplication of human chromosome 15q11-q13 region (Dup15q) has been strongly implicated in autism spectrum disorder (ASD). While the pathogenic role of paternally derived duplication remains uncertain. A recent genetic study has implicated a significant contribution of paternal duplication to ASD/DD/MCA with penetrance rate of ~20%. While the total number of subjects analyzed is large in this report but the number of paternal Dup15q case is still relatively small (~10). So the conclusion regarding the role of paternal Dup15q remains somewhat questionable but is stronger than previous data in literature.

The same group has reported the mutant mice carrying the large 6Mb duplication that mimic the common human 15q11-q13 duplication before. Unexpected, the paternal but not maternal duplication display some autism-like behaviors. It remains unclear why maternal duplication does not contribute any significant defect in mice. In this study, the author has produced a new line of smaller duplication (upstream of AS-IC and downstream of Ube3a within the homologous region of 15q11-q13. The paternal duplication of this smaller region did not result in the similar behavioral or cellular phenotypes of that observed in the large 6Mb duplication. It is noted that the same analysis of the small maternal duplication was not included in manuscript and could be valuable. Because of these results, the focus of the study was then turned to the paternally expressed genes proximal to the small duplicated region. A series of experiments including in vivo rescue has been conducted. The results provided the support that Ndn is the critical gene to responsible for the autism phenotypes of the large 6 Mb paternal duplication in mice. Overall, authors should be complimented on the large number of experiments that are both in vitro and in vivo and at cellular, functional and behavioral levels. The conclusion is reasonably convincing based on the results presented. However, several issues may be worthwhile to discuss in revision

We appreciate the thoughtful comments of this reviewer. For maternal duplication of 1.5 Mb Dp mice (1.5 Mb matDp/+ mice), we conducted behavioral tests according

to the reviewer's suggestion (new Supplementary Fig. 3). Although a slight increase in social interaction time was observed, this was differed from our original *15q dup* mice (i.e., less social interaction) and other behavioral tests could not detect abnormalities in *1.5 Mb matDp/+* mice. Thus, it is still difficult to answer why maternal duplication does not induce significant abnormalities in mice; however, our current data and a study from another laboratory (Smith et al., 2011) support the idea that the increased “single” copy number of *Ube3a* may not be sufficient to induce autistic-like phenotypes in mice. This issue is newly discussed in a "Limitations" section (revised manuscript line: 468-488).

Major concerns:

1. The reference #17 is cited as the major evidence supporting the pathogenicity of the paternal duplication of 15q11-q13. The large sample size was repeatedly mentioned. This statement may be misleading. The large population was screened for the 15q11-q13 CNV but the total number of cases for genotype and phenotype study of the paternal duplication is ~10 and small. In addition, the penetrance of paternal duplication is ~20% which indicate that many cases carrying the paternal duplications are not affected. This caveats of this report should be discussed in detail.

We thank this helpful comment. As the reviewers suggested, the study (Isles et al., 2016) provided screening of a large number of people but not a number of cases. We have corrected this issue in the Introduction as follows:

Line 91 (revised manuscript)

"To the contrary, however, an investigation with a large sample size found individuals with paternally derived duplication also met the criteria for ASD."

=>

"In contrast, an investigation found ... met the criteria for ASD, although its penetrance was estimated at ~20% and the number of cases is still small."

2. In figure 1 D, the level of the increased expression is Ube3a is less than other paternally expressed genes but it is statistically significant based on the data presented.

In addition, because the maternal allelic expression of Ube3a is cell type specific, the total RNAs from the brain tissue with mixed cell types may underestimate the difference too.

According to the reviewer's comment, we examined the expression level of *Ube3a* in primary hippocampal neurons to avoid mixed cell types. The substantial increase of *Ube3a* expression in brain tissue was not seen in primary hippocampal neurons (new Fig. 1e). Therefore, as the reviewer suggested, the increased *Ube3a* expression in 1.5Mb *patDp* mice is probably due to the expression in glial cells.

3. While the results from the 15q dup Δ Ndn mice provide reasonable evidence; because both behaviors and spine morphology are less specific as the readout for the manipulation, it seem that the more important rescuing experiments may be the neurochemical analysis of quantification of 5-HT and 5-HIAA in brain if technically feasible and at least electrophysiological recording experiment what should be feasible. Both of these were performed in 15q dup Δ Ndn mice and the large 6Mb duplication mice

According to the reviewer's important suggestion, we conducted the electrophysiological recording of 5-HT neurons in the dorsal raphe nuclei of 15q dup Δ Ndn mice. The original 15q dup mice showed hyperpolarization and decreased sEPSC amplitude in the DRN (Nakai et al., 2017). Interestingly, the resting membrane potential in 15q dup Δ Ndn mice showed a similar level to WT mice while sEPSC amplitude was tended to be decreased, which is similar to that of 15q dup mice (new Supplementary Fig. 9). The frequency of sEPSC was not altered as similar to 15q dup mice. Combined with our previous study that chronic administration of fluoxetine could rescue the sEPSC amplitude and not restore the resting membrane potential in the DRN of 15q dup mice, the investigation of a synergistic effect of 5-HT and *Ndn* may be necessary for further study. We added the result and edited the manuscript as follows:

Figure

Supplementary Fig. 9.

Result

Line 322-332 (revised manuscript)

Discussion

Line 432 (revised manuscript)

“Therefore, *Ndn* expression in DRN may play a key role in the development of autistic-like behaviors or aberrant intrinsic spine dynamics found in *15q dup* mice.”

=>

“In addition, we observed ameliorated resting membrane potential in the DRN of *15q dupΔNdn* mice (Supplementary Fig. S9). Furthermore, our previous study showed chronic administration of fluoxetine, a selective serotonin reuptake inhibitor, could normalize the amplitude of sEPSC in the DRN of *15q dup* mice⁹. Therefore, the synergistic effect of *Ndn* expression in DRN and 5-HT signaling may play a key role in the development of autistic-like behaviors or aberrant intrinsic spine dynamics found in *15q dup* mice.”

Material and Method

Line 731-762 (revised manuscript)

4. A similar restriction enzyme map that illustrates the genomic Southern analysis of 1.5 Mb duplication mice described in Figure 1 may help reader and future reference

According to the reviewer’s suggestion, we prepared the schema of a genomic map for generating 1.5 Mb duplication mice (new Supplementary Fig. 1).

5. Two different social behavior paradigms were used in this and previous paper. Only 3 chamber sociability test was used in the 6Mb paternal duplication mice.. Both reciprocal social interaction test and 3 chamber sociability test were used in mice with 15q dupΔNdn. The inconsistency among the experimental design may confound the data interpretation and should be discussed.

We apology this inconsistency. We have recently published that *15q dup* mice show decreased interaction time in the reciprocal social interaction test (Tsurugizawa et

al., 2020). Both 3 chamber sociability and reciprocal social interaction tests could detect abnormalities of *15q dup* mice. Therefore, we conducted these 2 behavioral analyses on *15q dup Δ Ndn* mice. To avoid confusion, we have edited the manuscript as below:

We added a paper of Tsurugizawa et al., 2020 (revised manuscript line: 294, ref23)

Line 293 (revised manuscript)

“decreased social interaction”

=>

“decreased social interaction (based on reciprocal social interaction and three-chambered social interaction tests)”

6. The interpretation and statement of E/I imbalance in text may be a bit premature because only the function of pyramidal neurons was measured and the E/I imbalance should be a net effect of E/I either at single cell or network level.

We agree with the reviewer's opinion that the E/I imbalance, often found in ASDs, is a net effect of excitatory and inhibitory neurons. We investigated E/I balance not only by electrophysiology but also by immunohistochemistry with anti-VGAT and VGLUT antibodies (Fig. 5f-h). As the reviewer indicated, our description regarding the overexpression of *Ndn* in our original manuscript was improper. In this revised manuscript, we avoided using "E/I balance" when we analyzed only pyramidal neurons by electrophysiology as below.

Line 241 (revised manuscript)

“Collectively, these findings indicate that increased expression of *Ndn* may contribute to the cortical E/I imbalance”

=>

“Collectively, these findings indicate that increased expression of *Ndn* contributes to the hyperexcitability in cortical pyramidal neurons.”

*7. Line 361: “We are the first to identify *Ndn* as a causative gene for Dup15q syndrome”.*

This may be an overstatement. Not sure the author has enough to conclude the NDN is the “causative” gene for human Dup15q syndrome based on the animal study. The conclusion for a causative role of a gene implicated in human should only come from human data. The animal and in vitro data may provide some evidence to support the case. Unless the phenotype is pathognomonic for human diseases, these evidence is not direct.

Thank you for the reviewer’s comment. As we replied to the comment of reviewer #2-8, we have corrected the indicated points to be more precise and also added a "Limitations" section in Discussion, indicating that this study is based on model animals (revised manuscript line: 468-488).

Line 353 (revised manuscript)

"We are the first to identify *Ndn* as a causative gene for Dup15q syndrome"

=>

" We are the first to identify *Ndn* as a critical gene in 15q11-q13 for developing of autistic-like behaviors in mice. "

Minor concerns

1. There are many discussion and comments in result sections. In general, these comments should be reserved in discussion section.

Thank you for this comment. We have edited following the reviewer’s suggestion.

Line 103 (revised manuscript)

“A previous study suggested that increased expression of *Snord115* (also named *MBII-52*) affected the activity of serotonin 2c receptor (5-HT2cR) via RNA editing and induced altered intracellular Ca²⁺ response elicited by a 5-HT2cR agonist in the primary cortical neurons in *15q dup* mice.”

=>

“A previous study suggested that increased expression of *Snord115* (also named *MBII-52*) affected the activity of serotonin 2c receptor (5-HT2cR) in *15q dup* mice.”

Line 185 (original manuscript)

“Ndn was reported to be expressed in postmitotic neurons²⁷. It interacts with many proteins including E2F1, p53, p75NTR or Maged1 and regulates downstream gene expression as a transcriptional cofactor, resulting in the regulation of cell cycle or apoptosis²⁸.”

=>removed

Line 319-326 (original manuscript)

“In summary, our screening revealed that one of the PEGs in the 15q11-q13...cortical circuits and prevention of autistic-like behaviors”

=>

moved to “Discussion” (revised manuscript line: 335-342)

2. Line 61: Autism spectrum disorder (ASD) is a neurodevelopmental disorder. ASD is not “a” disorder

We have corrected it as follows:

Line 61 (revised manuscript)

"Autism spectrum disorder (ASD) is a neurodevelopmental disorder."

=>

"Autism spectrum disorder (ASD) is referred to as a group of neurodevelopmental disorders."

3. Line 61-62: The prevalence rate of ASD is now estimated 1 in 59 according to the Centers for Disease Control (CDC). Add the reference for this statement

We have added the reference in this revised manuscript as shown below.

Baio, J. et al. Prevalence of Autism Spectrum Disorder Among Children Aged 8 Years — Autism and Developmental Disabilities Monitoring Network, 11 Sites, United States, 2014. MMWR. Surveill. Summ. 67, 1–23 (2018).

4. Line 64-65: “including rare variants of a single gene or pathogenic copy number variations (CNVs)”: rare variants of a single gene are also pathogenic

We have corrected as below:

Line 64 (revised manuscript)

“including rare variants of a single gene or pathogenic copy number variations (CNVs)”

=>

“including pathogenic rare variants of a single gene or copy number variations (CNVs)”

5. Line 66-68: There are reported association between CNVs and schizophrenia, but few evidence supporting the causality such as CNV in 15q11-q13. So be cautious to make a generalized statement

We agree to the reviewer’s comment and corrected the indicated sentence.

Line 65 (revised manuscript)

"Since both duplications and deletions of the same CNV can cause ASD and schizophrenia, the correlation between gene dosage and phenotypes is rather complex."

=>

"Some CNVs, including 15q11-q13 duplication, are overlapped as shared risk factors for both ASD and schizophrenia, suggesting that the correlation between gene dosage and phenotypes is rather complex"

6. One of the interesting and also intriguing questions remains why the maternal duplication of the region in the mice has largely not be able to recapitulate the human maternal duplication.

Although this issue is not our goal in this study and currently we do not have direct evidence, we propose the following hypothesis described in a “Limitations” section in this revised manuscript (revised manuscript line: 468-488).

Our study includes several concerns as follows. This study is limited to the animal.

Therefore, these results may not be direct evidence for explaining Dup15q etiology. There seems to be a different allelic contribution rate between humans and mice. Matthew Anderson's group reported intriguing results (Smith et al., 2011). They prepared 2 lines of BAC transgenic mice of *Ube3a*, an MEG. They named 1xTg and 2xTg, which had 2- and 3-fold increased expression of *Ube3a* compared to WT mice, respectively. Therefore, the expression of *Ube3a* is postulated as equivalent levels between 1xTg and our maternal *15q dup* mice. Similar to our results, the 1xTg mice did not show impaired social behavior while the 2xTg mice exhibited autistic-like behavior. Furthermore, subjects having interstitial Dup15q (2 maternal copies of *UBE3A*) often showed milder symptoms than isodicentric Dup15q (3 maternal copies of *UBE3A*) (Kaslner et al., 2015). From this evidence, mice may have higher resiliency for increased expression of *Ube3a* than that of humans, and the increased “single” copy number of *Ube3a* may not be sufficient to induce autistic-like phenotypes in mice. Compared to the MEGs, PEGs may have an opposite effect. In humans, paternal Dup15q is often normal (low penetrance); however, many abnormal phenotypes are found in mice from accumulated studies (Nakatani et al, 2009; Tamada et al, 2010; Ellegood et al., 2015; Kishimoto et al., 2015; Nakai et al, 2017; Nagano et al., 2018; Tsurugizawa et al., 2020; Septyaningtrias et al., 2020, etc). Thus, mice may have a higher susceptibility for the increased expression of PEGs than humans. We still do not know whether maternal and paternal Dup15q is an equivalent syndrome or not. Further investigation is required to clarify the difference between maternal and paternal Dup15q and the importance of paternal Dup15q.

Reviewers' Comments:

Reviewer #1:

Remarks to the Author:

The authors have addressed all my previous comments. The paper reads really well and is an important piece of work.

I spotted one minor error:

Line 473 "...an MEG' should be '...a MEG'

Reviewer #2:

Remarks to the Author:

The authors have improved this manuscript substantially, and it really is careful and helpful work for understanding the function of Ndn and for understanding ways in which the mouse models diverge from human disorders. There are still a few edits that will help ensure that these results are interpreted by the broader community in the right context. The general idea is that the authors should really specify that this is relevant to paternal dup15q mice, and not a mouse model for human dup15q. Specific comments are as follows:

1. The title should really reflect paternally-inherited dup15q. "Genetic dissection identifies Necdin as a driver gene in a mouse model of 15q duplication syndrome." should be "Genetic dissection identifies Necdin as a driver gene in a mouse model of paternal 15q duplications."

2. Throughout the manuscript, indicate paternal dup15q or maternal dup15q when referring to the mouse model. Examples are below, but this should be done throughout the manuscript:

Line 35 - In vivo genetic screening identified a critical gene, Necdin, for paternal 15q duplication syndrome in mice

Line 38 - Excision of a single copy Necdin from paternal 15q dup mice attenuated abnormal autistic...

Line 49—"driver gene for paternal dup15q in mice. We reveal that a hitherto underestimated gene in the context of paternal dup15q, Necdin (Ndn)"

Line 53—"generate paternal15q dupDNdn mice with a normalized copy number of Ndn by excising its one copy from paternal dup15q mice using..."

3. Minor text edits for accuracy as follows:

Line 37 - Necdin regulates formation and maturation of cortical dendritic spines. Add "in mice"

Line 45 - "...study has implicated a significant contribution of paternal duplication to ASD." Remove "significant", as it's not clear how significant this is.

Line 46—"etiology of maternally derived duplication is known to be UBE3A because this gene is...". Change "known" to "thought", unfortunately, this isn't known, yet.

Line 469—" Our study includes several concerns as follows. This study is limited to the animal. Therefore, these results may not be direct evidence for explaining Dup15q etiology. There seems to be a different allelic contribution rate between humans and mice."

" ("Our study has some important limitations. First, this study was carried out in mouse models, and there seems to be different contributions from the two parental alleles to the phenotypes observed in humans and mice. Therefore, these results may not be directly relevant for explaining Dup15q etiology in humans."

Line 482-483—"In humans, paternal Dup15q is often normal (low penetrance);" Actually paternal duplications of 15q11-q13 may have a distinct, somewhat mild disorder and perhaps a slightly increased risk for ASD. I don't think the studies are large enough to say that paternal duplication

causes low penetrance ASD.

Reviewer #3:

Remarks to the Author:

Authors have addressed this reviewer's comments and concerns fully.

We thank all the reviewers for positive feedback. Our responses to the points raised are as follows:

<Reviewer #1>

The authors have addressed all my previous comments. The paper reads really well and is an important piece of work. I spotted one minor error:

Line 473 "...an MEG' should be '...a MEG'

We appreciate the reviewer's comments. We corrected them as the reviewer indicated.

<Reviewer #2>

The authors have improved this manuscript substantially, and it really is careful and helpful work for understanding the function of Ndn and for understanding ways in which the mouse models diverge from human disorders. There are still a few edits that will help ensure that these results are interpreted by the broader community in the right context. The general idea is that the authors should really specify that this is relevant to paternal dup15q mice, and not a mouse model for human dup15q. Specific comments are as follows:

1. The title should really reflect paternally-inherited dup15q. "Genetic dissection identifies Necdin as a driver gene in a mouse model of 15q duplication syndrome." should be "Genetic dissection identifies Necdin as a driver gene in a mouse model of paternal 15q duplications."

We corrected it as the reviewer indicated.

2. Throughout the manuscript, indicate paternal dup15q or maternal dup15q when referring to the mouse model. Examples are below, but this should be done throughout the manuscript:

Line 35 - In vivo genetic screening identified a critical gene, Necdin, for paternal 15q duplication syndrome in mice

Line 38 - Excision of a single copy Necdin from paternal 15q dup mice attenuated abnormal autistic...

Line 49—“driver gene for paternal dup15q in mice. We reveal that a hitherto underestimated gene in the context of paternal dup15q, Necdin (Ndn)”

Line 53—“generate paternal15q dupDNdn mice with a normalized copy number of Ndn by excising its one copy from paternal dup15q mice using...”

We added “paternal” in the sentences as the reviewer indicated and throughout the manuscript as following points,

L73: multi-dimensional abnormalities in *15q dup* mice.

=> multi-dimensional abnormalities in **paternal** *15q dup* mice.

L93: altered spine dynamics found in *15q dup* mice.

=> altered spine dynamics found in **paternal** *15q dup* mice.

L94: excluding the target gene from *15q dup* mice.

=> excluding the target gene from **paternal** *15q dup* mice.

L100: serotonin 2c receptor (5-HT2cR) in *15q dup* mice.

=> serotonin 2c receptor (5-HT2cR) in **paternal** *15q dup* mice.

L133: this alteration differed from that of *15q dup* mice.

=> this alteration differed from that of **paternal** *15q dup* mice.

L140: show behavioral abnormalities found in *15q dup* mice...

=> show behavioral abnormalities found in **paternal** *15q dup* mice...

L226: pyramidal neurons in the original *15q dup* mice was...

=> pyramidal neurons in the original **paternal** *15q dup* mice was...

L236: characteristic is similar to that of neurons of *15q dup* mice shown...

=> characteristic is similar to that of neurons of **paternal** *15q dup* mice shown...

L243: removing a single copy of *Ndn* from original *15q dup* mice is...

=> removing a single copy of *Ndn* from original **paternal** *15q dup* mice is...

L250: fertilized eggs obtained by crossing *15q dup* male mice...

=> fertilized eggs obtained by crossing **paternal** *15q dup* male mice

L261: compared the gene expression level in the brain of *15q dupΔNdn* mice...

=> compared the gene expression level in the brain of **paternal** *15q dupΔNdn* mice...

L267: Single copy removal of *Ndn* from *15q dup* mice restores alteration in dendritic spines and cortical E/I imbalance found in *15q dup* mice

=> Single copy removal of *Ndn* from **paternal** *15q dup* mice restores alteration in dendritic spines and cortical E/I imbalance found in **paternal** *15q dup* mice

L270: spines in cortical neurons of *15q dupΔNdn*...

=> spines in cortical neurons of **paternal** *15q dupΔNdn*...

L272: enhanced spine formation rate in *15q dup* mice was completely abolished in *15q dupΔNdn* mice.

=> enhanced spine formation rate in **paternal** *15q dup* mice was completely abolished in **paternal** *15q dupΔNdn* mice.

L274: rate was also normalized in *15q dupΔNdn* mice (Fig. 5e).

=> rate was also normalized in **paternal** *15q dupΔNdn* mice (Fig. 5e).

L277: In our previous study, *15q dup* mice...

=> In our previous study, **paternal** *15q dup* mice...

L280: somatosensory cortex of *15q dup* mice (Fig. 5f and g). The number of

VGAT-immunopositive puncta in *15q dupΔNdn* mice was higher than that in *15q dup* mice (Fig. 5g),

=> somatosensory cortex of **paternal** *15q dup* mice (Fig. 5f and g). The number of VGAT-immunopositive puncta in **paternal** *15q dupΔNdn* mice was higher than that in **paternal** *15q dup* mice (Fig. 5g),

L287: *15q dupΔNdn* mice are recovered...

=> **Paternal** *15q dupΔNdn* mice are recovered...

L288: autistic-like behaviors found in *15q dup* mice,

=> **ASD**-like behaviors found in **paternal** *15q dup* mice,

L292: significant difference between WT and *15q dupΔNdn*...

=> significant difference between WT and **paternal** *15q dupΔNdn*...

L296: to verify the sociability for *15q dupΔNdn* mice...

=> to verify the sociability for **paternal** *15q dupΔNdn* mice...

L298: between *15q dupΔNdn* and WT mice (Fig. 6d). Supporting this finding, *15q dupΔNdn* mice...

=> between **paternal** *15q dupΔNdn* and WT mice (Fig. 6d). Supporting this finding, **paternal** *15q dupΔNdn* mice...

L303: other quadrants both in WT and *15q dupΔNdn* mice...

=> other quadrants both in WT and **paternal** *15q dupΔNdn* mice...

L307: autistic-like behaviors in *15q dup* mice.

=> **ASD**-like behaviors in **paternal** *15q dup* mice.

L309: To reveal the communication alteration in *15q dupΔNdn* mice,

=> To reveal the communication alteration in **paternal** *15q dupΔNdn* mice,

L312: Based on our previous studies, *15q dup* mice emitted...

=> Based on our previous studies, **paternal** *15q dup* mice emitted...

L314: Both *15q dup* and *15q dupΔNdn* pups exhibited more...

=> Both **paternal** *15q dup* and *15q dupΔNdn* pups exhibited more...

L315: Moreover, *15q dupΔNdn* adult male mice...

=> Moreover, **paternal** *15q dupΔNdn* adult male mice...

L318: social communication found in *15q dup* mice...

=> social communication found in **paternal** *15q dup* mice...

L321: raphe nuclei of *15q dupΔNdn* mice. Our previous study revealed *15q dup* mice...

=> raphe nuclei of **paternal** *15q dupΔNdn* mice. Our previous study revealed **paternal** *15q dup* mice...

L324: This analysis revealed that *15q dupΔNdn* mice...

=> This analysis revealed that **paternal** *15q dupΔNdn* mice...

L326: DRN (vmDRN), similar to *15q dup* mice...

=> DRN (vmDRN), similar to **paternal** *15q dup* mice...

L327: frequency of sEPSC in *15q dupΔNdn* mice...

=> frequency of sEPSC in **paternal** *15q dupΔNdn* mice...

L329: membrane potential found in *15q dup* mice was not observed in *15q dupΔNdn* mice (Supplementary Fig. 9e and f), suggesting the rescued intrinsic excitability of 5-HT neurons in *15q dupΔNdn* mice.

=> membrane potential found in **paternal** *15q dup* mice was not observed in **paternal** *15q dupΔNdn* mice (Supplementary Fig. 9e and f), suggesting the rescued intrinsic excitability of 5-HT neurons in **paternal** *15q dupΔNdn* mice.

L335: phenotypes seen in *15q dup* mice.

=> phenotypes seen in **paternal** *15q dup* mice.

L337: reminiscent of the phenotypes in *15q dup* mice.

=> reminiscent of the phenotypes in **paternal** *15q dup* mice.

L339: multi-dimensional abnormalities found in *15q dup* mice.

=> multi-dimensional abnormalities found in **paternal** *15q dup* mice.

L354: increased expression of *Snord115* in *15q dup* mice...

=> increased expression of *Snord115* in **paternal** *15q dup* mice...

L359: abnormalities found in original *15q dup* mice...

=> abnormalities found in original **paternal** *15q dup* mice...

L372: Surprisingly, *15q dupΔNdn* mice...

=> Surprisingly, **paternal** *15q dupΔNdn* mice...

L369: several impaired behaviors found in *15q dup* mice.

=> several impaired behaviors found in **paternal** *15q dup* mice.

L394: support of this idea, *15q dup* mice...

=> support of this idea, **paternal** *15q dup* mice...

L400: cortical E/I balance in *15q dup* mice was...

=> cortical E/I balance in **paternal** *15q dup* mice was...

L406: restoring only *Ndn* from *15q dup* mice...

=> restoring only *Ndn* from **paternal** *15q dup* mice...

L432: membrane potential in the DRN of *15q dupΔNdn* mice...

=> membrane potential in the DRN of **paternal** *15q dupΔNdn* mice...

L434: sEPSC in the DRN of *15q dup* mice⁹.

=> sEPSC in the DRN of **paternal** *15q dup* mice⁹.

L437: intrinsic spine dynamics found in *15q dup* mice.

=> intrinsic spine dynamics found in **paternal** *15q dup* mice.

L439: is the decreased 5-HT seen in *15q dup* mice.

=> is the decreased 5-HT seen in **paternal** *15q dup* mice.

L443: *Ndn* via a 5-HT-independent pathway in *15q dupΔNdn* mice.

=> *Ndn* via a 5-HT-independent pathway in **paternal** *15q dupΔNdn* mice.

L450: developing cortex is altered in *15q dup* and recovered in *15q dupΔNdn* mice.

=> developing cortex is altered in **paternal** *15q dup* and recovered in **paternal** *15q dupΔNdn* mice.

L459: early treatment (P3-P21) of SSRI to *15q dup* mice alleviated...

=> early treatment (P3-P21) of SSRI to **paternal** *15q dup* mice alleviated...

L461: The increased expression of *Ndn* in *15q dup* mice...

=> The increased expression of *Ndn* in **paternal** *15q dup* mice...

L491: like phenotypes found in *15q dup* mice.

=> like phenotypes found in **paternal** *15q dup* mice.

L494: normalizing the copy number of *Ndn* in *15q dup* mice...

=> normalizing the copy number of *Ndn* in **paternal** *15q dup* mice...

L837: Behavioral analysis in *15q dupΔNdn* mice

=> Behavioral analysis in **paternal** *15q dupΔNdn* mice

L1124: abnormal behaviors found in *15q dup* mice

=> abnormal behaviors found in **paternal** *15q dup* mice

L1126: A schematic of phenotypes of *15q dup* mice

=> A schematic of phenotypes of **paternal** *15q dup* mice

L1180: cortical excitatory/inhibitory imbalance found in *15q dup* mice

=> cortical excitatory/inhibitory imbalance found in **paternal** *15q dup* mice

L1184: expression in the frontal cortex of *15q dup* and *15q dupΔNdn*...

=> expression in the frontal cortex of **paternal** *15q dup* and *15q dupΔNdn*...

L1203: Figure 6. *15q dupΔNdn* mice are recovered from abnormal behaviors found in *15q dup* mice

=> Figure 6. **Paternal** *15q dupΔNdn* mice are recovered from abnormal behaviors found in **paternal** *15q dup* mice

Supplementary Figure 8. Altered social communications in *15q dup* mice were retained in *15q dupΔNdn* mice

=> Supplementary Figure 8. Altered social communications in **paternal** *15q dup* mice were retained in **paternal** *15q dupΔNdn* mice

Supplementary Figure 9. Electrophysiological analyses of *15q dupΔNdn* mice in the dorsal raphe nucleus

(a-d) Comparisons between WT and *15q dupΔNdn* mice for sEPSC amplitude...

=> Supplementary Figure 9. Electrophysiological analyses of **paternal** *15q dupΔNdn* mice in the dorsal raphe nucleus

(a-d) Comparisons between WT and **paternal** *15q dupΔNdn* mice for sEPSC amplitude

3. *Minor text edits for accuracy as follows:*

Line 37 - Necdin regulates formation and maturation of cortical dendritic spines. Add "in mice"

Line 45 – "...study has implicated a significant contribution of paternal duplication to ASD." Remove "significant", as it's not clear how significant this is.

Line 46—"etiology of maternally derived duplication is known to be UBE3A because this gene is...". Change "known" to "thought", unfortunately, this isn't known, yet.

Line 469—" Our study includes several concerns as follows. This study is limited to the animal. Therefore, these results may not be direct evidence for explaining Dup15q etiology. There seems to be a different allelic contribution rate between humans and mice."

" □ "Our study has some important limitations. First, this study was carried out in mouse models, and there seems to be different contributions from the two parental alleles to the phenotypes observed in humans and mice. Therefore, these results may not be directly relevant for explaining Dup15q etiology in humans."

We corrected above 4 comments as the reviewer indicated.

Line 482-483—"In humans, paternal Dup15q is often normal (low penetrance);" Actually paternal duplications of 15q11-q13 may have a distinct, somewhat mild

disorder and perhaps a slightly increased risk for ASD. I don't think the studies are large enough to say that paternal duplication causes low penetrance ASD.

According to the reviewer's comment, we modified this sentence as follows :

“In humans, paternal Dup15q is often normal (low penetrance), however,...”

=>

“In humans, paternal Dup15q is often normal (low penetrance and/or milder symptoms than that of maternal Dup15q), however,...”

<Reviewer #3>

Authors have addressed this reviewer's comments and concerns fully.

We appreciate the reviewer's comments for improving our manuscript.